# Federated Learning via Meta-Variational Dropout

**Insu Jeon**[*]    **Minui Hong**[†]    **Junhyeog Yun**[*]    **Gunhee Kim**[†]
Seoul National University, Seoul, South Korea
[*]{insuj3on,antemrdm}@gmail.com
[†]{alsdml123,gunhee}@snu.ac.kr

## Abstract

Federated Learning (FL) aims to train a global inference model from remotely distributed clients, gaining popularity due to its benefit of improving data privacy. However, traditional FL often faces challenges in practical applications, including model overfitting and divergent local models due to limited and non-IID data among clients. To address these issues, we introduce a novel Bayesian meta-learning approach called meta-variational dropout (MetaVD). MetaVD learns to predict client-dependent dropout rates via a shared hypernetwork, enabling effective model personalization of FL algorithms in limited non-IID data settings. We also emphasize the posterior adaptation view of meta-learning and the posterior aggregation view of Bayesian FL via the conditional dropout posterior. We conducted extensive experiments on various sparse and non-IID FL datasets. MetaVD demonstrated excellent classification accuracy and uncertainty calibration performance, especially for out-of-distribution (OOD) clients. MetaVD compresses the local model parameters needed for each client, mitigating model overfitting and reducing communication costs. Code is available at https://github.com/insujeon/MetaVD.

## 1 Introduction

Federated learning (FL) aims at training a global model from distributed clients without sharing or collecting their sensitive raw data. Thanks to its privacy-preserving aspect of FL [1], it is increasingly popular to be applied to various applications such as image classification [2, 3], object detection [4, 5], keyboard suggestion [6, 7], recommendation [8, 9], and healthcare [10, 11]. The conventional FL algorithm could achieve convergence when the data from different clients is independently and identically distributed (IID) [12–14]. However, due to the differences in preferences, locations, and usage habits of clients, the private data in FL are usually non-IID. When the data distributions of the clients vary, the local model learned from each client can diverge, and thus learning an optimal global model could fail [15, 16]. Furthermore, the scale of client data may not be sufficient to train a local model with large parameters, causing model overfitting and poor generalization [17, 18].

To overcome the challenge caused by non-IID data, personalized federated learning (PFL) has emerged [15, 16]. In the PFL, each client is allowed to have its own personalized model trained on each client's local data while still participating in the global model training. There are many branches of PFL, such as those based on multitask learning [19, 20], meta-learning [21–26], and transfer learning [27–30]. Although these approaches improve training convergence in non-IID data settings, they may still experience model overfitting with limited client data. Recently, the Bayesian learning paradigm has also been introduced in FL to address the overfitting by incorporating uncertainty in the model parameters [31–37]. However, they may also struggle with divergent local models when the data from different clients exhibit significant statistical variability. Motivated by these challenges, we aim to address the issues of FL with the non-IID and limited client data simultaneously.

In this paper, we present Meta-Variational Dropout (MetaVD), a novel Bayesian meta-learning approach [38–42] developed for PFL. MetaVD learns to predict client-dependent dropout rates

37th Conference on Neural Information Processing Systems (NeurIPS 2023).

via a hypernetwork [25, 43–45]. This mechanism facilitates a data-efficient estimation of client-specific posterior simply by modulating a shared global NN parameter across all clients. The adaptation of MetaVD to conventional FL algorithms (e.g., FedAvg [1], Reptile [22], MAML [21], or PerFedAvg [23]) allows flexible model personalization across a variety of non-IID client distributions. MetaVD also incorporates a unique model aggregation strategy based on client-specific dropout uncertainty, providing a principled Bayesian way to consolidate local models into a global model. This strategy significantly enhances the convergence of the FL algorithm on non-IID data. In addition, MetaVD inherits the merit of Variational Dropout (VD) [46, 47], which facilitates model compression. This not only improves model generalization for out-of-data (OOD) clients but also reduces communication costs for parameter exchange. We performed an extensive analysis on MetaVD covering a wide range of FL scenarios [18], including different scales, non-IID degrees, partitions, and multi-domain settings [48]. In all of these experiments, MetaVD achieves excellent results compared to other state-of-the-art baselines.

## 2 Background

A standard approach to FL (*e.g.,* FedAvg [1]) iterates between the local training on the client devices and global optimization at the server. Given $M$ clients, each of which has a data set $\mathcal{D}^m = \{(x_i^m, y_i^m)\}_{i=1}^{|\mathcal{D}^m|}$, the FL problem can be formulated as follows:

$$\text{Server: } \min_w \ \mathcal{J}(w) = \sum_{m=1}^{M} g^m \mathcal{J}^m(w), \quad \text{Client: } \mathcal{J}^m(w) = \frac{1}{|\mathcal{D}^m|} \sum_i \ell(x_i^m, y_i^m; w). \quad (1)$$

$w$ denotes the model parameter, and the global learning objective $\mathcal{J}(w)$ at the server is a weighted average of the local objectives $\mathcal{J}^m(w)$ over $M$ clients. The weight $g^m$ is proportional to the size of the local dataset (e.g., $|\mathcal{D}^m|/|\mathcal{D}|$). The local loss in a client device is usually defined as the empirical negative log-likelihood on the $m$-th client's dataset $\mathcal{D}^m$ (i.e., $\ell(\mathcal{D}^m; w) = -\log p(y^m|x^m, w)$).

The local training is carried out in parallel fully (or partly) in each client device, with multiple Stochastic Gradient Descent (SGD) [49] epochs to get the fine-tuned local parameter $w^m$. Then, the aggregation step computes the global parameter in the server by taking the weighted average of the local parameters (i.e., $\bar{w} \leftarrow \sum_{m=1}^{M} g^m w^m$). The global model parameter $\bar{w}$ is then used as the initial parameter for each client in the next round of local training. FL aims to train models on large distributed datasets by only exchanging model parameters (e.g., $\bar{w}$ and $w^m$) between server and local devices, thereby minimizing privacy leakage of client data.

**Challenges in FL.** There are many challenges to real-world FL applications. (1) *Heterogeneity of client data*. The original FL algorithm converges well when the client data is IID [12–14]. However, the client data often has different characteristics (e.g., classes or tasks follow non-IID). The $w^m$ would drift away from each other, causing the $\bar{w}$ to be suboptimal [15, 16]. (2) *Sparse participation*. In practice, the total number of clients $M$ can be extremely large, while communication between the server and the clients can be intermittent or unreliable. This creates the challenge of inconsistent training due to a small subset of participating clients in each round of communication [18]. (3) *Poor generalization due to limited data*. When the training data available on each local device is limited, the local model can easily overfit, resulting in poor generalization to unseen clients [17, 18]. (4) *Communication cost*. FL optimization requires frequent communication between local devices and the central server to exchange model parameters. This process is slow and could introduce additional privacy concerns. Therefore, reducing model size is also an important area of research [50].

**Bayesian FL.** While conventional FL methods use a point estimate of the parameter as in Eq.1, recent work [31–33] has incorporated probability distributions over the model parameters. In these Bayesian FL approaches, the client device first estimates the local posterior from its data, and then the server aggregates the partially updated local posteriors into a global posterior. Based on FedAvg, a Gaussian distribution is used to represent each parameter in FedAG [32], and a posterior aggregation strategy using the MCMC technique is proposed in FedPA [33]. FedBE [31] uses a Bayesian ensemble global model with Gaussian or Dirichlet distributions. These methods improve prediction confidence and model convergence. However, FedAG [32] and FedBE [31] assume a global Gaussian posterior, but only consider point estimates of local parameters. FedPA [33] uses global and local Gaussian posteriors, but only maintains the global posterior mean on the server, making it difficult to track local uncertainties. In addition, the performance of previous methods can still be compromised by the

statistical discrepancies in the client data, as they are not tailored to the heterogeneous FL scenario. Recently, Bayesian PFL methods have also been introduced for non-IID data [34–37, 51, 52]. For example, pFedGP [34] learns a common Gaussian kernel for all clients and infers personalized classifiers for each client. pFedBayes [35] assumes an independent posterior model for each device while learning a shared global prior. These approaches achieve a substantial predictive performance improvement in non-IID data senerios compared to the previous Bayesian FL approaches. However, their approach still has limitations, such as the high computational cost of inverting a large kernel matrix in pFedGP [34] and imposing strong constraints in pFedBayes [35]. Their model aggregation rule does not account for parameter uncertainty directly. In addition, probabilistic modeling typically requires additional parameters to approximate the density, which can increase communication costs.

## 3 Approach: Meta-Variational Dropout

We propose a new Bayesian PFL approach, called Meta-Variational Dropout (MetaVD), which can simultaneously handle model personalization, regularization, and compression. MetaVD also exploits the uncertainties in model aggregation, thereby improving the training convergence on non-IID data. In addition, MetaVD is a universal approach that is compatible with various existing FL algorithms.

### 3.1 Variational Inference for FL

Instead of approximating the local posterior using MCMC in Bayesian FL [31–33], we can utilize the (amortized) Variational Inference (VI) framework of Bayesian meta-learning [38–42] that is originally developed for few-shot multi-task learning [53–55]. Considering each $m$-th client in FL as an individual task, an evidence lower-bound (ELBO) $\mathcal{L}_{\text{ELBO}}$ over all the $M$ datasets is defined as

$$\max_{\phi} \mathcal{L}_{\text{ELBO}}(\phi) = \sum_{m=1}^{M} g^m \{\mathbb{E}_{q(w^m;\phi)}[\log p(y^m|x^m, w^m)] - \text{KL}(q(w^m;\phi)||p(w^m))\}. \quad (2)$$

Here, $p(y^m|x^m, w^m)$ represents a likelihood model constructed using a neural network (NN) on each $m$-th client's data [56–59], and $w^m$ is a client-specific NN parameter. $q(w^m;\phi)$ is a variational posterior (or probabilistic) distribution over the $w^m$, which is also characterized by $\phi$. $g^m$ is the local weight as defined in Eq.(1). $p(w^m)$ is a prior distribution that acts as a regularizer for the $q(w^m;\phi)$. The local ELBO defined on each $m$-th client in Eq.(2) makes tradeoffs between the expected log-likelihood on its local dataset $\mathcal{D}^m$ and the KL divergence with a prior. In fact, maximizing the $\mathcal{L}_{\text{ELBO}}(\phi)$ with respect to the variational parameter $\phi$ is equivalent to minimizing $\sum_{t=1}^{M} g^m \text{KL}(q(w^m;\phi)||p(w^m|\mathcal{D}^m))$. In theory, $q(w^m;\phi)$ is trained to approximate the true local posterior distribution over the $w^m$ (i.e., $q(w^m;\phi) \approx p(w^m|\mathcal{D}^m)$) [56–59].

A straightforward Bayesian PFL approach might be utilizing a separate local posterior model $q(w^m;\phi)$ for each client device (e.g., , $\phi = (\phi^1, \cdots, \phi^M)$). However, only a sparse subset of clients can participate in each FL round due to the communication availability of edge devices. Moreover, the number of clients $M$ can be extremely large in practice, and some clients might only have small datasets. Thus, learning the variational parameter $\phi^m$ independently for each device is difficult. As an alternative, we introduce a new hypernetwork-based [25, 43–45] conditional dropout posterior modeling approach that can be data-efficiently trained across multiple clients in FL.

### 3.2 Meta-Variational Dropout

**Posterior model.** To promote efficient model personalization and reduce overfitting in Bayesian FL, we define the posterior model in Eq.2 based on a Variational Dropout (VD) technique that multiplies continuous Gaussian noise to the NN parameters during training to prevent overfitting [46, 47, 60–62]. MetaVD extends the VD posterior by employing a global hypernetwork that learns to predict client-specific dropout rates (or personal model structure). In MetaVD, the variational posterior distribution for each $m$-th client's parameter, $q(w^m;\phi)$ in Eq.2, can be constructed as

$$q(w^m;\phi = (\theta, \psi, e^m)) = \prod_{k=1}^{K} \mathcal{N}(w_k^m|\theta_k, \alpha_k^m \theta_k^2) \text{ where } \alpha^m = h_\psi(e^m). \quad (3)$$

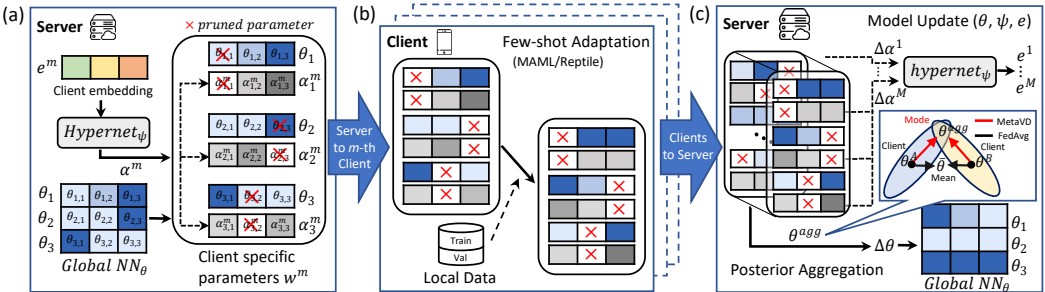

Figure 1: Overview of the Meta-Variational Dropout algorithm. (a) The server's hypernetwork predicts client-specific dropout rates from client embedding, $e^m$. The global parameters, $\theta$, and dropout variables, $\alpha^m$, are sent to each $m$-th client. (b) A few-shot local adaptation is performed on each client's device in parallel; then, the updated parameters are sent back to the server. (c) The server then aggregates those parameters to update its variational parameters $\theta$, $\psi$, and $e$.

The variational parameter $\phi$ is characterized by three distinct parameters $(\theta, \psi, e)$. The $\theta$ is a global NN model parameter kept at the server (with no client index $m$), where $K$ is the size of the model parameter. The posterior distribution over the $m$-th client's weight $w^m$ is described as a product of independent Gaussian distributions. In the product term in Eq.(3), each $k$-th factor describes a conditional Gaussian noise multiplication to the global NN parameter (e.g., $w^m = \theta * \epsilon^m$ where $\epsilon^m \sim \mathcal{N}(\vec{1}, \alpha^m)$ and $\vec{1}$ is an $K$-dimensional all-one vector). The $\alpha_k^m$ represents the client-specific dropout variable[1] on each $k$-th index of NN parameter $\theta_k$. The $h_\psi$ indicates a hypernetwork parameterized by $\psi$ [43–45, 45] predicts the client-specific dropout rate $\alpha^m$. The $e^m$ is a learnable client embedding used as an input to the $h_\psi$. In essence, MetaVD is a technique that modulates a global NN parameter with Gaussian noises. By predicting the client-specific dropout variable via a hypernetwork, a single NN can be reconfigured for various different clients. Since learning a hypernetwork across multiple local clients is more efficient than learning all the local posteriors independently, this approach can mitigate the sparse client participation and limited data issues in FL.

**Prior model**. To optimize the posterior model $q(w^m; \phi)$, we need to specify the KL divergence term and the prior model $p(w^m)$ in Eq.(2). Prior modeling in MetaVD requires two criteria: (i) the KL divergence terms must be independent of the NN parameter $\theta$ to ensure the lower-bound assumption in VD [46, 47, 61, 62], and (ii) all clients must share the same prior model to support the multiplicative posterior aggregation rule in Bayesian FL. We adopt the hierarchical prior [64, 65, 47] discussed in [47] because of its straightforward analytic KL term derivation and proven effectiveness in network regularization and sparsification. Under the hierarchical prior assumption, the KL divergence term in Eq.(2) is simplified to $\mathrm{KL}(q(w^m; \phi)||p(w^m)) = \sum_{k=1}^K 0.5 \log(1 + (\alpha_k^m)^{-1})$; see Appendix B for a more detailed derivation. This KL term is independent of the global NN parameter $\theta$ and efficiently regularizes the dropout variable. The same hierarchical prior is applied to all $M$ clients.

**Client-side objective.** Initially, the hypernetwork in the server approximates the dropout variable $\alpha^m$ for each $m$-th client. The global parameter $\theta$ and $\alpha^m$ are transmitted to each client device. Then, the posterior model in a $m$-th client is trained on local data using the following ELBO objective:

$$\max_{\theta, \alpha^m} \mathcal{L}_{\mathrm{ELBO}}^m(\theta, \alpha) = \frac{1}{|\mathcal{D}^m|} \sum_i \log p(y_i^m | x_i^m, f(\epsilon; \theta, \alpha^m)) - \sum_{k=1}^K 0.5 \log(1 + (\alpha_k^m)^{-1}), \quad (4)$$

which maximizes $\mathcal{L}_{\mathrm{ELBO}}^m$ on the client dataset $\mathcal{D}^m = \{(x_i^m, y_i^m)\}_{i=1}^{|\mathcal{D}^m|}$ with respect to the variational parameters (i.e., $\theta$, $\alpha^m$). The optimization can be done by the *stochastic gradient variational Bayes* (SGVB). [56–58], which reparameterizes the random weight variable $w^m$ using a differentiable transformation as $w_k^m = f(\epsilon_k; \theta_k, \alpha_k^m) = \theta_k + \sqrt{\alpha_k^m} \theta_k \epsilon_k$. with a random IID noise $\epsilon_k \sim \mathcal{N}(0, 1)$. The intermediate weight $w_k^m$ is now differentiable with respect to $\theta_k$, hence $\alpha_k^m$, and can be optimized via the SGD [49]. The analytical KL term acts as a regularization for $\alpha_k^m$.

**Server-side objective.** Once a local posterior adaptation is done in each client device, the updated parameters $\theta_*^m$ and $\alpha_*^m$ are returned to the server, then the server updates the variational parameters

---

[1]The dropout rate is $p = \alpha/(1 + \alpha) \in [0, 1]$ [63, 46]. We refer to $\alpha$ as the dropout variable for simplicity.

**Algorithm 1** MetaVD algorithm with MAML and Reptile variant for FL

---

**Input:** # of communication round $R$, # of client $N$, server learning rate $\eta$, client learning rate $\gamma$, inner learning rate $l$, local steps $E$, inner steps $I$, KL divergence parameter $\beta$.

Initialize a global parameter $\theta$, hypernetwork $h_\psi(\cdot)$ and $e$.

**for** $r = 1, ..., R$ **do**          ▷ FL Rounds
    Sample $M$ clients from $1, ..., N$ clients
    **for** $m = 1, ..., M$ **do**
        Set $\theta^m = \theta$ and $\alpha^m = h_\psi(e^m)$
        Send $\theta^m$ and $\alpha^m$ to the $m$-th client
        $\theta_*^m, \alpha_*^m \leftarrow$ LOCALADAPTATION$(\theta^m, \alpha^m)$
    Compute global param $\theta_*^{\text{agg}}$ using $\theta_*^m, \alpha_*^m$ and Eq.(5)
    $\Delta\theta \leftarrow \theta_*^{\text{agg}} - \theta$
    $\Delta\alpha^m \leftarrow \alpha_*^m - \alpha^m$
    $\theta \leftarrow \theta + \eta\Delta\theta$
    $\psi \leftarrow \psi + \eta\frac{1}{M}\Sigma_m g_m((\nabla_\psi\alpha^m)^T\Delta\alpha^m)$
    **for** $m = 1, \cdots, M$ **do**
        $e^m \leftarrow e^m + \eta(\nabla_{e^m}\psi)^T(\nabla_\psi\alpha^m)^T\Delta\alpha^m$

**procedure** LOCALADAPTATION_MAML$(\theta, \alpha)$
    **for** Each local step $i$ from 1 to $E$ **do**
        Set $\theta_i' = \theta_i$ and $\alpha_i' = \alpha_i$
        Sample dataset $D_{\text{tr}}^m$ and $D_{\text{val}}^m$ from $D^m$
        **for** Each inner step $j$ from 1 to $I$ **do**
            $\theta_i' \leftarrow \theta_i' - l\nabla_{\theta_i'}\mathcal{L}_{ELBO}^m(\theta_i', \alpha_i'; D_{\text{tr}}^m))$
        $\theta_i \leftarrow \theta_i - \gamma\nabla_{\theta_i}\mathcal{L}_{ELBO}^m(\theta_i', \alpha_i'; D_{\text{val}}^m)$
        $\alpha_i \leftarrow \alpha_i - \gamma\nabla_{\alpha_i}\mathcal{L}_{ELBO}^m(\theta_i', \alpha_i'; D_{\text{val}}^m)$
    Set $\theta_* = \theta_i$ and $\alpha_* = \alpha_i$
    Send $\theta_*$ and $\alpha_*$ to the server

**procedure** LOCALADAPTATION_REPTILE$(\theta, \alpha)$
    **for** Each local step $i$ from 1 to $E$ **do**
        $\theta_i \leftarrow \theta_i - \gamma\nabla_{\theta_i}\mathcal{L}_{ELBO}^m(\theta_i, \alpha_i; D^m))$
        $\alpha_i \leftarrow \alpha_i - \gamma\nabla_{\alpha_i}\mathcal{L}_{ELBO}^m(\theta_i, \alpha_i; D^m)$
    Set $\theta_* = \theta_i$ and $\alpha_* = \alpha_i$
    Send $\theta_*$ and $\alpha_*$ to the server

---

(e.g., $\theta, \psi, e^m$) using the updated and local parameters. To update the global NN parameter $\theta$, we first assume the Bayesian posterior aggregation rule: $p(w|\mathcal{D}) \propto \prod_{m=1}^M p(w^m|\mathcal{D}^m)$ [31–33, 66]. Since each local posterior in Eq.(3) is a Gaussian (dropout) distribution, their product is also Gaussian: $\mathcal{N}(w|\theta_*^{\text{agg}}, \cdot) \approx \prod_{m=1}^M \mathcal{N}(w^m|\theta_*^m, \alpha_*^m(\theta_*^m)^2)$. This gives us an exact aggregation rule[2] to compute the maximum a posterior (MAP) solution of the $\theta_*^{\text{agg}}$ as follows:

$$\theta_*^{\text{agg}} = \frac{1}{M}\sum_m r^m\theta_*^m \quad \text{where} \quad r^m = \frac{g^m(\alpha_*^m(\theta_*^m)^2)^{-1}}{\sum_m g^m(\alpha_*^m(\theta_*^m)^2)^{-1}}. \tag{5}$$

Note that Eq.(5) has the intuitive interpretation that the aggregation weight $r^m$ is inversely proportional to its corresponding dropout variable (or noise variance) $\alpha_*^m$. Thus, parameters with high uncertainty have correspondingly less influence on the global mean prediction. In this way, we can fully exploit the uncertainty of the model parameters in the aggregation. To fully update each global NN parameter $\theta_k$, we follow the similar parameter update rule of FedAvg, except that the Bayesian aggregation Eq.(5) is utilized as in Algo.1. For the parameter of hypernetwork $\psi$, a more general update rule is used following [25]. We compute the changes in the updated dropout for each client $\Delta\alpha^m$ as described in Algo.1, then an approximated gradient of the hypernetwork parameter is computed using the chain rule as $\nabla_\psi\mathcal{L}_{\text{ELBO}}^m(\alpha^m) = (\nabla_\psi\alpha^m)^T\Delta\alpha^m$, where $\nabla_\psi\alpha^m$ is the gradient of the output of hypernetwork with respect to $\psi$. $\Delta\alpha^m$ is an approximation of the vector-jacobian product, which we are inspired by the work of [25] and [69]. The gradient for $\nabla_{e^m}\mathcal{L}_{\text{ELBO}}^m(\alpha^m)$ can be derived using the similar chain rule. The detailed update rules for each parameter are in Algo.1.

**Combination with other meta-learning algorithms.** Since the KL regularization in Eq.(4) is independent of the global (or initial) parameter $\theta$, MetaVD can be combined with several existing meta-learning based PFL algorithms (e.g., Reptile [22, 70], MAML [21, 55], PerFedAvg [23, 71]). For example, MAML [21, 55] requires some internal update steps using a subsampled dataset to compute the second-order gradient for $\theta$. Reptile [22, 70] uses only a first-order gradient computation as described in Algo.1, which is similar to FedAvg except for the addition of the learning rate $\eta$. Each local adaptation step is performed using the $\mathcal{L}_{\text{ELBO}}^m$ in Eq.(4) and the local data set $D^m$. Unlike conventional meta-learning based PFL algorithms, which maintain only one global initialization parameter, MetaVD allows to change the mode of the initialization parameters for each client.

## 4 Experiments

To validate the MetaVD approach, we conducted extensive experiments in various scenarios following the FL benchmark research [18], including different degrees of non-IID and client participation rates.

---

[2]We adapted this rule from [33]. In general, the mode of the product of $M$ Gaussians, $\prod_{m=1}^M \mathcal{N}(\mu^m, \sigma^m)$, simplifies to $\mu^{\text{agg}} = \sum_{m=1}^M r^m\mu^m$ where $r^m = ((\sigma^m)^2)^{-1}/\sum_{m=1}^M((\sigma^m)^2)^{-1}$ [67]. Interestingly, Eq.(5) is equivalent to maximizing the logarithm of the product of the weighted posteriors [68]. In the heterogeneous data, the product rule can achieve a smaller aggregation error than the mixture of Gaussian posteriors.

| Hetrogenity | *CIFAR-100* dataset | | | | | | *CIFAR-10* dataset | | |
| | $\dot{\alpha} = 5.0$ | | | $\dot{\alpha} = 0.5$ | | | $\dot{\alpha} = 0.1$ | | |
| Method | Test (%) | OOD (%) | Δ | Test (%) | OOD (%) | Δ | Test (%) | OOD (%) | Δ |
|---|---|---|---|---|---|---|---|---|---|
| FedAvg [1] | 42.35 | 43.08 | +0.73 | 41.92 | 41.96 | +0.04 | 71.65 | 71.57 | −0.08 |
| FedAvg+FT [18] | 41.49 | 42.45 | +0.96 | 40.99 | 39.83 | −1.16 | 69.62 | 68.38 | −1.24 |
| FedProx [73] | 42.23 | 44.11 | +1.88 | 42.03 | 40.51 | −1.52 | 72.27 | 73.75 | +1.48 |
| FedBE [31] | 45.17 | 45.43 | +0.26 | 44.29 | 44.23 | −0.06 | 70.23 | 69.19 | −1.04 |
| pFedGP [34] | 42.69 | 43.07 | +0.38 | 42.44 | 42.53 | +0.09 | 71.94 | 76.83 | +4.89 |
| Reptile [22] | 47.87 | 47.73 | −0.14 | 46.13 | 45.94 | −0.19 | 73.93 | 76.36 | +2.43 |
| MAML [21] | 48.30 | 49.14 | +0.84 | 46.33 | 46.65 | **+0.32** | 76.06 | 74.89 | −1.17 |
| PerFedAvg (HF-MAML) [23] | 48.19 | 47.35 | −0.84 | 46.22 | 46.36 | +0.14 | 75.42 | 79.56 | +4.14 |
| FedAvg+MetaVD (ours) | 47.82 | 50.26 | **+2.44** | 47.54 | 47.55 | +0.01 | 76.87 | 76.25 | −0.62 |
| Reptile+MetaVD (ours) | **53.71** | **54.50** | +0.79 | **52.06** | **51.50** | −0.56 | 76.51 | **82.07** | +5.56 |
| MAML+MetaVD (ours) | 52.40 | 51.78 | −0.62 | 50.21 | 49.75 | −0.46 | **77.27** | 79.05 | +1.78 |
| PerFedAvg+MetaVD (ours) | 51.67 | 51.70 | +0.03 | 50.02 | 48.70 | −1.32 | 76.06 | 81.77 | **+5.71** |

Table 1: Classification accuracies with different (non-IID) heterogeneity degrees of $\dot{\alpha} = [5.0, 0.5]$ in CIFAR-100 and $\dot{\alpha} = 0.1$ in CIFAR-10. The higher score, the better.

We used multiple FL datasets [72], including CIFAR-10, CIFAR-100, FEMINIST, and CelebA. To evaluate the effectiveness of the hypernetwork, we also performed an ablation study comparing MetaVD to regular VD. Additionally, we assessed the uncertainty calibration and model compression ability of MetaVD. Finally, we test MetaVD with multi-domain datasets.

**Baselines.** We compare our method with standard FL methods such as FedAvg [1] and FedProx [73], meta-learning PFL algorithms like Reptile [22], MAML [21], and PerFedAvg [23], and Bayesian FL methods including FedBE [31] and pFedGP [34]. To ensure consistency, we employ the widely-used CNN model for FL [3, 74, 25] across all baselines. Additionally, "fine-tuning" (FT) refers to performing few-shot adaptation steps with FedAvg before evaluation. An overview of baselines can be found in Appendix D.

**Implementation.** For reproducibility, we run experiments in a containerized environment that simulates FL communication with clients only on a server. We test $T = 1000$ of total FL rounds, following the conventions in [18]. One baseline model can be run on a single GPU. All experiments are run on a cluster of 32 NVIDIA GTX 1080 GPUs. MetaVD's hypernetwork consists of an embedding layer of dimension $(1 + M/4)$, followed by three fully connected NNs with a Reaky ReLU activation and an exponential activation for the dropout logit output. The predicted dropout variable is then applied to the global weight of other baselines. In our study, we apply MetaVD to only one fully connected layer before the output layer [75–78], which leads to significant performance improvements in all experiments. See Appendix E for implementation details. Our code is available at https://github.com/insujeon/MetaVD.

## 4.1 Generalization on Non-IID Settings

**Datasets and training.** To evaluate the generalization capabilities of the models under non-IID data conditions, we perform tests on both CIFAR-10 and CIFAR-100 datasets with varying degrees of heterogeneity. We follow the similar evaluation protocols of the pFL-Bench [18], using Dirichlet allocation to partition each dataset into 130 clients with different Dirichlet parameters, denoted as $\dot{\alpha} = [5, 0.5, 0.1]$. As shown in Figure 2, the class labels and data size per client are heterogeneous across clients. A smaller $\dot{\alpha}$ represents a higher degree of heterogeneity. To evaluate

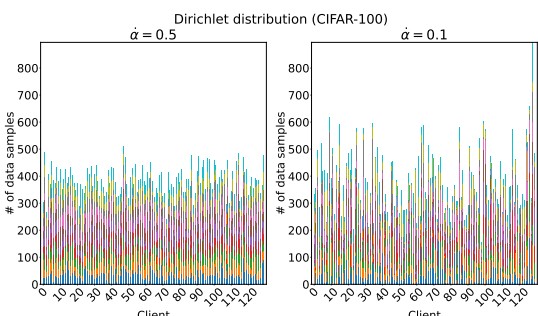

Figure 2: Visualization of client's data distribution in different non-IID degrees ($\dot{\alpha} = [0.5, 0.1]$).

the test accuracy and generalization performance of the baselines on new clients, we randomly select 30 out of 130 clients as out-of-distribution (OOD) clients, which are not involved in the training phase. See Appendix G for more details.

**Results.** Table 1 shows the weighted average classification accuracy for participating (Test) and non-participating (OOD) clients on CIFAR-10 and CIFAR-100 datasets with varying non-IID degrees. The generalization gap, denoted by $\Delta$, represents the difference between OOD and Test accuracy. As shown in Table 1, PFL methods such as Reptile, MAML, and PerFedAvg generally outperform non-PFL methods such as FedAvg and FedProx. While the Bayesian ensemble approach, FedBE, improves on FedAvg, it still lags behind PFL methods in OOD accuracy. Additionally, pFedGP exhibits suboptimal performance in the presence of heterogeneous distribution of data among clients. As $\dot{\alpha}$ changes from $5.0$ to $0.5$, the test and OOD accuracy of all models decreases as the degree of non-IID increases. When combined with MetaVD, all baselines show significant performance improvements, regardless of whether they are FL or PFL algorithms (e.g., Reptile enhances from $47.87$ to $53.71$ adapting MetaVD).. These results demonstrate the adaptability and effectiveness of MetaVD in mitigating model overfitting and handling non-IID client data in FL contexts.

## 4.2 Ablation Study

**Settings.** To evaluate the capability of the hypernetwork in MetaVD, we perform an ablation study by comparing MetaVD with naive VD [46] and EnsembleVD [79] approaches on the CIFAR-100 dataset. The naive (global) VD model maintains a global dropout parameter shared by all clients; the dropout parameter is treated as a global model parameter, as in FedAvg. EnsembleVD maintains $M$ independent dropout parameters for all clients. The client-specific dropout parameter can be stored in each client analogous to the partial FL [80, 81, 5]. In contrast, MetaVD utilizes a hypernetwork to learn the personal dropout rate across all clients. Bayesian posterior aggregation rules is applied in all model based on dropout rates to update the global model parameter [82, 83].

**Results.** Table 2 outlines the results of the ablation study; MetaVD's hypernetwork-based posterior modeling outperforms all other baselines. The dropout rates in baselines such as VD or EnsembleVD could not fully learn the independent dropout variables well due to restricted client participation. On the other hand, both the hypernetwork and the global parameter converge well in MetaVD. This observation demonstrates that MetaVD's hypernetwork provides a more data-efficient approach to learning client-specific model uncertainty compared to other baselines. Further ablation studies performed on the FEMNIST dataset are shown in Appendix H.

| *CIFAR-100* dataset | | | |
|---|---|---|---|
| **Method** | Test (%) | OOD (%) | $\Delta$ |
| Reptile [22] | 47.87 | 47.73 | $-0.14$ |
| Reptile+VD [46] | 50.20 | 49.28 | $-0.92$ |
| Reptile+EnsembleVD [79] | 52.49 | 52.36 | $-0.13$ |
| Reptile+MetaVD (ours) | **53.71** | **54.50** | $+\mathbf{0.79}$ |

Table 2: MetaVD ablation study in CIFAR-100.

## 4.3 Uncertainty Calibration

| *CIFAR-100* dataset | | |
|---|---|---|
| **Method** | ECE (%) | MCE (%) |
| FedAvg [1] | 0.60 | 36.79 |
| FedAvg+FT [18] | 0.69 | 45.04 |
| FedProx [73] | 0.67 | 39.69 |
| FedBE [31] | 0.50 | 34.66 |
| Reptile [22] | 0.77 | 50.52 |
| MAML [21] | 0.75 | 46.57 |
| PerFedAvg (HF-MAML) [23] | 0.69 | 45.27 |
| FedAvg+MetaVD (ours) | **0.39** | **25.27** |
| Reptile+MetaVD (ours) | 0.57 | 42.40 |
| MAML+MetaVD (ours) | 0.52 | 37.26 |
| PerFedAvg+MetaVD (ours) | 0.43 | 30.20 |

Table 3: Uncertainty calibration scores (ECE and MCE) measured on the OOD client in CIFAR-100 ($\dot{\alpha} = 0.1$). The lower, the better.

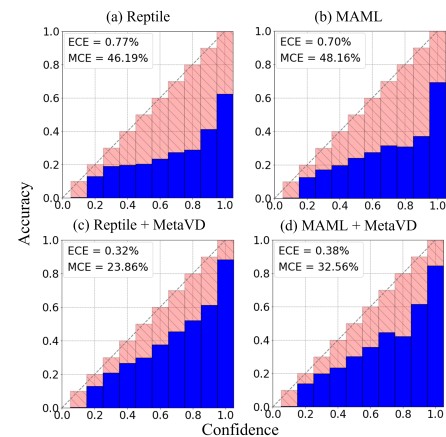

Figure 3: Reliability diagrams for (a) Reptile, (b) MAML, (c) Reptile+MetaVD, and (d) MAML+MetaVD in CIFAR-100.

**Settings.** Identifying any potential bias in the model's prediction is important to avoid serious consequences, especially when the model is used to make important decisions [84–86]. In the FL

environment, where customers have limited non-IID data, it is even more critical to properly calibrate the prediction model. Therefore, we investigate whether the proposed MetaVD approach can also improve the calibration measures for FL baselines. The Expected Calibration Error (ECE) measures the expected deviation between a model's predicted probability and the actual positive class frequency, while the Maximum Calibration Error (MCE) measures the maximum difference. These calibration metrics are commonly used to evaluate the reliability of probabilistic predictions.

**Results.** Table 3 summarizes the ECE and MCE values tested with the OOD clients in the CIFAR-100 dataset and shows that the meta-learning based PFL algorithms (e.g., Reptile, MAML, and PerFedAvg) tend to have higher ECE and MCE values than the conventional FL algorithms (e.g., FedAvg, FedProx, and FedBE). This means that PFL baselines achieve high classification accuracy, but their probability predictions are more likely to be biased. This may be a byproduct of the additional optimization-based adaptation steps of meta-learning with limited local client data. On the other hand, our MetaVD approach significantly reduces both ECE and MCE values for all meta-learning-based methods, indicating that MetaVD effectively mitigates overfitting and reduces bias on the OOD clients. Figure 3 shows the reliability diagrams that visualize the model calibration [84–86] in CIFAR-100 ($\dot{\alpha} = 0.5$). They plot the expected sample accuracy as a function of confidence. If the model is perfectly calibrated, then the diagram becomes the identity function. Any deviation from a perfect diagonal represents miscalibration. While Reptile and MAML tend to be overconfident in their predictions, Reptile+MetaVD comes fairly close to the desired diagonal function. Remarkably, in most instances, adapting MetaVD leads to improved model calibration.

### 4.4 Client Participation

|  | *FEMNIST* dataset | | | | | | | | |
|---|---|---|---|---|---|---|---|---|---|
| **Sparsity** | $s = 0.2$ | | | $s = 0.1$ | | | $s = 0.05$ | | |
| **Method** | Test (%) | OOD (%) | $\Delta$ | Test (%) | OOD (%) | $\Delta$ | Test (%) | OOD (%) | $\Delta$ |
| FedAvg [1] | 88.08 | 85.29 | −2.79 | 88.13 | 84.70 | −3.43 | 88.06 | 86.22 | −1.84 |
| FedAvg+FT [18] | 88.33 | 86.37 | −1.96 | 87.85 | 86.95 | −**0.90** | 87.66 | 87.11 | −0.55 |
| Reptile [22] | 88.55 | 86.52 | −2.03 | 88.39 | 87.20 | −1.19 | 87.86 | 88.22 | +**0.36** |
| FedAvg+MetaVD (ours) | 88.81 | 85.28 | −3.53 | 88.69 | 85.71 | −2.98 | 88.66 | 86.21 | −2.45 |
| Reptile+MetaVD (ours) | **89.90** | **89.04** | −**0.86** | **89.86** | **88.63** | −1.23 | **89.43** | **88.71** | −0.72 |

Table 4: Results of classification accuracies with different participant client rates of $s = 0.2$, $s = 0.1$, and $s = 0.05$ in the FEMNIST dataset. We report the results of participating clients (Test) and non-participating clients (OOD). The higher the better.

**Settings.** In real-world FL scenarios, such as intermittent connections between clients and servers or limited client device performance, numerous clients may be unable to participate in each FL round. This is important for cross-device FL with a large number of clients or resource-limited clients. In this experiment, we evaluated the performance of the methods under different levels of client participation in each FL round. We experimented with 200 clients in the FEMNIST dataset. For each FL round, we randomly selected 40, 20, and 10 clients to participate during training, with participating client rates $s$ of 0.2, 0.1, and 0.05, respectively. To measure OOD accuracy, we excluded 40 preselected clients from the selection of 200 clients so that they do not participate in the entire training.

**Results.** The overall classification results with different participant client rates $s$ are summarized in table 4. Reptile performs better than FedAvg. The effect of data heterogeneity on performance degradation becomes more severe as more clients participate in training. The decrease in test accuracy as the participating client rate $s$ becomes smaller is due to having less training data as fewer clients participate in each round. Meanwhile, Reptile+MetaVD outperforms the other baselines. Interestingly, the performance drop for Reptile+MetaVD is not as significant as for Reptile, showing that MetaVD can adapt well to FL scenarios with smaller participant sizes.

### 4.5 Muti-domain Datasets

**Settings.** Existing federated learning algorithms usually assume a single-domain approach, where only one dataset is used in the experiment. Multi-domain learning [87–89] aims to utilize all available training data across different domains to improve the performance of the model. In this section, we

| Method | 2 Domains (a) | | 2 Domains (b) | | 2 Domains (c) | | 3 Domains (d) | |
|---|---|---|---|---|---|---|---|---|
| | Test (%) | OOD (%) | Test (%) | OOD (%) | Test (%) | OOD (%) | Test (%) | OOD (%) |
| FedAvg [1] | 43.65 | 43.45 | 64.02 | 52.88 | 86.81 | 81.01 | 63.73 | 55.55 |
| Reptile [22] | 48.92 | 48.93 | 66.13 | 56.22 | 87.50 | 83.05 | 66.98 | 57.12 |
| MAML [21] | 47.39 | 48.51 | 66.56 | 55.72 | 88.57 | 84.52 | 67.08 | 58.15 |
| PerFedAvg (HF-MAML) [23] | 49.21 | 50.91 | 66.57 | 55.24 | 87.94 | 83.75 | 67.20 | 57.03 |
| FedAvg+MetaVD (ours) | 48.23 | 48.62 | 65.58 | 56.85 | 87.38 | 82.67 | 65.93 | 58.58 |
| Reptile+MetaVD (ours) | **52.26** | **54.75** | **68.35** | 59.07 | **88.63** | 85.03 | 68.78 | 61.59 |
| MAML+MetaVD (ours) | 51.34 | 52.82 | 68.24 | 61.21 | 88.59 | 85.02 | **68.81** | **61.60** |
| PerFedAvg+MetaVD (ours) | 51.18 | 53.06 | 67.93 | **61.32** | 88.27 | **85.32** | 68.05 | 61.17 |

Table 5: Classification accuracies with multi-domain datasets. (a) CelebA + CIFAR-100, (b) CIFAR-100 + FEMNIST, (c) CelebA + FEMNIST, (d) CelebA + CIFAR-100 + FEMNIST.

further evaluate the performance of PFL algorithms on a multi-domain FL dataset, where we assume that each client can have data from different domains. We use three different FL datasets to construct the multi-domain task distributions: FEMNIST, CIFAR-100, and CelebA. To sample each client's local data, we use the Dirichlet sampling technique ($\dot{\alpha} = 0.5$) used in the §4.1.

**Results.** Table 5 shows the classification accuracies for the Test and OOD clients when using a combination of two or three datasets. The meta-learning based PFL algorithm consistently outperforms FedAvg in terms of classification accuracy. In addition, when MetaVD is applied to either the FL or PFL algorithms, it significantly improves prediction accuracy in all multi-domain settings. Notably, MetaVD shows greater improvements in OOD accuracy compared to the improvement in test accuracy. Overall, these results demonstrate the versatility of MetaVD in improving robustness and generalization even in multi-domain FL datasets.

## 4.6 Model Compression

**Settings.** Federated learning optimization requires frequent communication of model parameters between devices and the central server, which can be slow and may raise privacy concerns. Therefore, minimizing communication costs by reducing the size of model parameters is an important issue in FL. To explore the compression capabilities of MetaVD, we performed an additional experiment on the CIFAR-10 dataset. The sign DP indicates that each model parameter is dropped during the FL communication.

| *CIFAR-10* dataset ($\dot{\alpha} = 0.5$) | | | |
|---|---|---|---|
| Method | Test (%) | OOD (%) | Sparsity(%) |
| Reptile+MetaVD | **83.20** | **83.40** | 0 |
| MAML+MetaVD | 81.32 | 81.81 | 0 |
| PerFedAvg+MetaVD | 81.06 | 81.47 | 0 |
| Reptile+MetaVD+DP | 81.40 | 80.98 | **80.06** |
| MAML+MetaVD+DP | 81.48 | 81.73 | 79.49 |
| PerFedAvg+MetaVD+DP | 82.43 | 82.19 | 78.20 |

Table 6: Results of model compression. MetaVD+DP does not communicate the model parameters whose dropout rates are larger than $0.8$.

We used the thresholding technique to drop the model parameter; Any parameter with a dropout rate greater than $0.8$ was dropped during the FL rounds.

**Results.** Table 6 shows the test and OOD accuracy results, as well as the sparsity (%) in the CIFAR-100 dataset. The sparsity represents the proportion of zero-value model parameters in the personalized layer. A higher sparsity percentage indicates more parameter pruning or elimination performed on the weight. In our experiment, MetaVD was able to prune about $80\%$ of the weights in the personalized layer. In addition, when we dropped the communication of the parameters between the client and the server using the dropout thresholding technique, MetaVD still showed relatively good performance. In the case of Reptile+MetaVD+DP, the performance decreased by about $2\%$ while using only $20\%$ of the weights. On the other hand, MAML+MetaVD+DP and PerFedAvg+MetaVD+DP show an improvement in performance of about $1\%$. This shows that MetaVD can compress the model parameters required in the personalized layer, reducing the communication cost in FL without sacrificing much performance. Appendix K shows more experiments on model compression results on the CIFAR-100 dataset with different non-IID settings.

# 5 Conclusion

In this study, we presented a new novel Bayesian personalized federated learning (PFL) approach called meta-variational dropout (MetaVD). MetaVD utilizes a hypernetwork that predicts the dropout rates for each independent NN parameter, which enables effective model personalization and adaptation in federated learning (FL) with non-IID and limited data scenarios. In addition, MetaVD is the first approach to exploit variational dropout uncertainty in posterior aggregation in PFL. MetaVD's dropout posterior modeling provides a principled Bayesian aggregation strategy to consolidate local models into a global model, thereby improving training convergence. MetaVD is also a generic approach that is compatible with any other existing meta-learning-based PFL algorithms to avoid model overfitting. In addition, MetaVD's ability to compress model parameters can be used to reduce communication costs. A potential limitation of our approach is that it may increase the complexity of the model by introducing an additional hypernetwork. However, the hypernetwork is kept on the server, and we have verified that applying MetaVD to just one last layer before the output layer yields significant performance improvements in all experiments. Experimentally, MetaVD's performance has been validated on several PFL benchmarks, including CIFAR-10, CIFAR-100, FEMINIST, and CelebA, as well as multi-domain datasets. It demonstrates superior classification accuracy and uncertainty calibration, especially for out-of-distribution (OOD) clients. Overall, the experimental results show MetaVD to be a highly versatile approach capable of addressing many challenges in FL.

## Acknowledgment

This work was supported by Center for Applied Research in Artificial Intelligence(CARAI) grant funded by Defense Acquisition Program Administration(DAPA) and Agency for Defense Development(ADD) (UD190031RD), Institute of Information & communications Technology Planning & Evaluation (IITP) grant (No.2019-0-01082, SW StarLab and No.2021-0-01343, Artificial Intelligence Graduate School Program (Seoul National University)), Basic Science Research Program through National Research Foundation of Korea (NRF) funded by the Korea government (MSIT) (NRF-2020R1A2B5B03095585). Gunhee Kim is the corresponding author.

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

# A  The Global Posterior Decomposition in Bayesian FL

In Bayesian estimation, an alternative to Maximum Likelihood Estimation (MLE) is the inference or estimation of the posterior distribution of the parameters given all the data, denoted as $p(w|\mathcal{D} \equiv \mathcal{D}^1 \cup \cdots \cup \mathcal{D}^M)$. This posterior distribution is proportional to the product of the likelihoods and a prior, $p(w|\mathcal{D}) \propto p(w) \prod_{m=1}^M p(\mathcal{D}^m|w)$. In the case of a uniform prior, the modes of the global posterior coincide with the MLE solutions. This establishes an equivalence between the inference of the posterior mode and optimization. Under the uniform prior, any global posterior distribution that exists decomposes into a product of local posteriors:

$$P(\theta|\mathcal{D}) \propto \prod_{m=1}^M p(\theta|\mathcal{D}^m) \tag{6}$$

This proposition is well discussed in the recent Bayesian FL work [33]. In the FL context, the global model can be computed in the server by multiplicatively aggregating the local models adapted to each client. However, in practice, aggregating the posterior with the overfitted local models into the global one is difficult due to the heterogeneity among clients' data and the permutation invariance property of the NN architecture [90]. Thus, frequent communications between the server and the client are still demanded in the FL. The challenge of inferencing the local and global model and improving communication efficiency remains an active research area for real federated applications.

# B  The Motivation of Hierarchical Prior in MetaVD

In Section 3.2, we adopt the hierarchical prior [64, 65, 91] proposed in [47] for several reasons. The first reason is simply that it is a well-posed Bayesian prior that can avoid a degenerate posterior problem of the conventional VD prior [46, 61, 62]. The second reason is that we want a sparse prior to reduce communication costs by compressing the model; the hierarchical prior has been proven effective for parameter pruning. Although we briefly mentioned the KL divergence term as $\mathrm{KL}(q(w^m; \phi)||p(w^m)) = \sum_{k=1}^K 0.5 \log(1 + (\alpha_k^m)^{-1})$ in the manuscript for readability, here we provide more detailed descriptions of applying the hierarchical prior.

Under the hierarchical prior assumption, we consider the joint prior and joint posterior distributions [92]. The joint prior, $p(w^m, \gamma^m) = p(w^m|\gamma^m)p(\gamma^m)$, is defined as a combination of a zero-mean Gaussian distribution, $p(w^m|\gamma^m) = \mathcal{N}(w^m|0, \gamma^m)$, and a uniform hyper-prior, $p(\gamma^m) = \mathcal{U}(\gamma^m|a, b)$, over the variance. Then, we define a (conditional) joint variational posterior, $q(w^m, \gamma^m|\phi) = q(w^m|\phi)q(\gamma^m)$, comprising the (conditional) dropout posterior $q(w^m; \phi = (\theta, \phi, e^m))$ defined in Section 3.2 of the manuscript and an additional Dirac delta distribution, $q(\gamma^m)$, to approximate the true (joint) posterior $p(w^m, \gamma^m|\mathcal{D}^m)$ (given the client's dataset $\mathcal{D}^m$).

**ELBO.** With the hierarchical prior $p(w^m, \gamma^m) = p(w^m|\gamma^m)p(\gamma^m)$ and the (conditional) joint posterior $q(w^m, \gamma^m|\phi) = q(w^m|\phi)q(\gamma^m)$ of the MetaVD, we can derive the local objective for each client as follows:

$$\mathrm{KL}(q(w^m, \gamma^m|\phi)||p(w^m, \gamma^m|\mathcal{D}^m)) = \int q(w^m, \gamma^m|\phi) \log \frac{q(w^m, \gamma^m|\phi)}{p(w^m, \gamma^m|x^m, y^m)} \partial w^m \partial \gamma^m$$

$$= \int q(w^m, \gamma^m|\phi) \log \frac{q(w^m, \gamma^m|\phi)p(y^m|x^m)}{p(y^m|x^m, w^m)p(w^m, \gamma^m)} \partial w^m \partial \gamma^m \tag{7}$$

$$= \int q(w^m, \gamma^m|\phi) \left\{ \log \frac{q(w^m, \gamma^m|\phi)}{p(w^m, \gamma^m)} + \log p(y^m|x^m) - \log p(y^m|x^m, w^m) \right\} \partial w^m \partial \gamma^m$$

$$= \mathrm{KL}(q(w^m, \gamma^m|\phi)||p(w^m, \gamma^m)) + \log p(y^m|x^m) - \mathbb{E}_{q(w^m, \gamma^m|\phi)}[\log p(y^m|x^m, w^m)]. \tag{8}$$

Eq. (2) is derived from Bayes' rule: $p(w^m, \gamma^m|\mathcal{D}^m) = \frac{p(y^m|x^m, w^m)p(w^m, \gamma^m)}{p(y^m|x^m)}$. By reordering Eq.(3), we get

$$\log p(y^m|x^m) \geq \mathbb{E}_{q(w^m, \gamma^m|\phi)}[\log p(y^m|x^m, w^m)] - \mathrm{KL}(q(w^m, \gamma^m|\phi)||p(w^m, \gamma^m)) \tag{9}$$

$$= \mathbb{E}_{q(w^m|\phi)}\mathbb{E}_{q(\gamma^m)}[\log p(y^m|x^m, w^m)] - \mathrm{KL}(q(w^m, \gamma^m|\phi)||p(w^m, \gamma^m))$$

$$= \mathbb{E}_{q(w^m|\phi)}[\log p(y^m|x^m, w^m)] - \mathrm{KL}(q(w^m, \gamma^m|\phi)||p(w^m, \gamma^m)) \tag{10}$$

The lower-bound in Eq.(4) is due to the positivity of the $\mathrm{KL}(q(w^m, \gamma^m|\phi)||p(w^m, \gamma^m|\mathcal{D}^m))$. Here, Eq. (5) corresponds to the ELBO objective of Eq. (3) in the manuscript.

**KL term.** If we further decompose the KL divergence term in Eq. (5),

$$\mathrm{KL}(q(w^m, \gamma^m|\phi)||p(w^m, \gamma^m)) = \mathrm{KL}(q(w^m|\phi)||p(w^m|\gamma^m)) + \mathrm{KL}(q(\gamma^m)||p(\gamma^m))$$

$$= \sum_{k=1}^K \{0.5 \log(1 + (\alpha_k^m)^{-1})\} + \sum_{k=1}^K \{\log(b - a)\} \tag{11}$$

The $\log(b - a)$ is independent of the unknown variables $\alpha^m$, $\theta$, and $\gamma^m$. Thus, we do not need to specify the value of hyperparameters $a$ and $b$ and can neglect them in practice[3]. Eq. (6) provides the rationale behind the

---

[3]For a more detailed proof of this, please see the appendix section of [47]

KL divergence term that we mentioned in Section 3.2 of the manuscript. In fact, the independence between the KL term and the parameter $\theta$ ensures compatibility with other optimization meta-learning algorithms. Also, the two-level structure in a hierarchical system can generate a much more complex distribution, expanding the potential solution spaces for selecting feasible prior. Thus, the hierarchical prior is a suitable prior for interpreting the variety of different clients' models in the FL environment. The same hierarchical prior is uniformly applied across all $1...M$ clients to ensure the global posterior decomposition assumption in Eq.(1).

## C    Additional Related Works

**Bayesian Neural Networks and Variational Dropout.**    To address the issue of overfitting caused by limited data, Bayesian neural networks (BNN) [93, 56, 94, 57] were proposed, which impose a prior distribution on each parameter. Various inference approaches have been proposed to model the posterior distribution of BNNs [58, 46, 59]. Variational dropout (VD) [60, 46, 61, 62, 47] encompasses a set of techniques that model the variational posterior distribution based on the dropout regularization method. Dropout regularization [95, 96] randomly deactivates some of the model parameters during training by multiplying them with discrete Bernoulli random noise. This method was initially popularized to prevent overfitting in neural network models. Subsequently, fast dropout [63, 97] demonstrated that multiplying the parameters with continuous noise sampled from Gaussian distributions yields similar results to conventional Bernoulli dropout [95]. VD approaches often employ sparse priors for regularization [46, 61, 62, 47], which facilitate the learning of independent dropout rates on the neural network parameters as variational parameters. This property distinguishes VD approaches from conventional dropout, which uses a single fixed rate for all parameters.

## D    Dataset and Methods

We follow the datasets and the evaluation protocol of pFL-Bench [18], which is a recently proposed benchmark for federated learning.

**Datasets.**    Here, we present descriptions of the dataset used in our experiment.

- The **CIFAR-10** and **CIFAR-100** datasets [98] are popular for 10-class and 100-class image classification respectively. Each dataset contains 50,000 training and 10,000 test images with a resolution of 32x32 pixels. Following the heterogeneous partition manners used in [18], we use Dirichlet allocation to split this dataset into 130 clients with different Dirichlet parameters as $\dot{\alpha} = [5, 0.5, 0.1]$ (a smaller $\dot{\alpha}$ indicates a higher heterogeneous degree).

- The Federated Extended MNIST (**FEMNIST**) is a widely used FL dataset for 62-class handwritten character recognition [72]. The original FEMNIST dataset contains 3,550 clients and each client corresponds to a character writer from EMNIST [99]. Following [18], we adopt the sub-sampled version, which contains 400 clients and a total of 85350 training and 21536 test images with a resolution of 28x28 pixels.

- The **CelebA** is a FL dataset based on [100] for 2-class image classification; Smiling or Not. Following [18], we adopt the sub-sampled version, which contains 500 clients and a total of 8752 training and 2347 test images with a resolution of 84x84 pixels. Each client is assigned images of a single celebrity.

We randomly select 30 clients for CIFAR-10 and CIFAR-100 and 40 clients for FEMNIST as OOD clients who do not participate in the FL processes. For all of the datasets, we follow the same heterogeneous patterns exhibited in the pFL-Bench, which covers a wide range of scales, partition manners, and non-IID degrees. This enables comprehensive comparisons and analysis among different methods in the non-IID data environment.

**Baselines.**    We present an overview of the baseline models used in our experiment, covering various popular and state-of-the-art approaches across three categories: Non-PFL, meta-learning-based PFL, and Bayesian FL methods.

The following **Non-PFL methods** are considered in our experiments:

- FedAvg [1] is a standard FL algorithm that averages gradients weighted by the data size of clients in each FL round.

- FedProx [73] employs a proximal term to encourage updated local models for clients not to deviate too much from the global model.

The following (meta-learning-based) **PFL methods** are considered in our experiment:

- Reptile [22] inserts a meta-learning fine-tuning phase of [70, 55] after the federated averaging algorithm stage to provide a reliable personalized model.

- MAML [21] enhances federated learning by integrating a MAML-based meta-learner [55], dividing the dataset into support and query sets for more robust local training.

- PerFedAvg [23] method focuses on the convergence analysis of the HF-MAML algorithm [71] in the FL scenario, offering a provably convergent method based on MAML to tackle non-convex functions.

The following **Bayesian FL** algorithm are also compared in our experiment:

- FedBE [31] enhances robust aggregation by adopting a Bayesian inference approach, sampling high-quality global models, and combining them through Bayesian model ensemble using Gaussian or Dirichlet distributions fitted to local models.

- pFedGP [34] employs Gaussian processes for personalized federated learning, leveraging shared kernel functions parameterized by a neural network and a novel inducing point approach for the generalization in the low data regime.

The following **Pruning** algorithm is also compared in our compression experiment in the appendix:

- SNIP [101, 102] algorithm is a deep learning pruning technique that identifies and removes less important connections in neural networks before the training begins using a small subset of the dataset. It is compared with our algorithm in the model compression experiment in the FL environment.

Additionally, Fine-tuning (FT) in the baseline's name indicates fine-tuning the local models with a few steps before evaluation within the FL processes, which is similar to the adaptation step of the optimization-based meta-learning.

## E    Implementation Details.

**Models.**    To maintain consistency with previous research, we employ the widely adopted CNN model for all algorithms and baselines [25, 74, 3]. Specifically, the global model comprises three convolutional layers with 64 filters and 3x3 kernels, followed by three fully-connected layers of 256, 128, and 64 hidden units.

The hypernetwork architecture of MetaVD consists of an embedding layer, followed by two consecutive blocks containing a linear layer and a LeakyReLU activation function, and one block containing a linear layer with exponential activation to output the dropout logit parameter $\alpha$. The dimension of client embedding $e^m$ is proportional to the number of clients $M$ and is calculated as $(1 + M/4)$. The hidden units' size in the hypernetwork was set to 200. The predicted dropout logit parameter is then applied to each weight of the MetaVD layer within the global model. In our study, we selectively apply MetaVD to just one fully-connected layer right before the output layer of the global model. This simple adaptation of MetaVD only in one fully-connected layer yielded significant performance improvements across all experiments.

**Hyperparameters.**    For all datasets, we set $T$ to 1000 to ensure sufficient convergence following conventions [18]. The batch size was set to 64, and local steps was set to 5. Personalization was executed with a batch size of 64 and a 1-step update. In order to ensure a fair comparison between the algorithms, the results presented in all of our experiments are obtained with the optimal hyperparameters for each model. To do so, we conducted an extensive parameter optimization using an optimization tool called Optuna[4] [103]. We employed both the Tree-structured Parzen Estimator algorithm and Random Sampler as hyperparameter samplers in Optuna.

For all methods, we investigated the server learning rate and local SGD learning rate within identical ranges. The server learning rate $\eta$ was explored within the range of $[0.6, 0.7, 0.8, 0.9, 1.0]$. The local SGD learning rate was investigated within the range of $[0.005, 0.01, 0.015, 0.02, 0.025, 0.03]$. In MAML and PerFedAvg, an additional client learning rate $\gamma$ is required, for which we searched within the range of $[0.01, 0.02, 0.03, 0.04, 0.05, 0.06, 0.07, 0.08, 0.09, 0.1]$. For MetaVD, an additional KL divergence parameter $\beta$ is needed, and we sought its optimal value within the range of $[1, 2, 3, 4, 5, 6, 7, 8, 9, 10, 11, 12, 13, 14, 15]$. We follow the hyperparameter setting outlined in pFedGP, except for adjusting the batch size to 64 or 320 and investigating the learning rate within the range of [0.03 to 0.1]. To ensure the reproducibility of the experiments, we will release all code, including baselines, on our GitHub repository.

## F    Experiment Outline

To evaluate the effectiveness and robustness of the proposed PFL methods, we tested the algorithms under various FL scenarios, including:

---

[4]https://optuna.org/

- Degree of Data Heterogeneity: we assessed the performance of each algorithm in scenarios where data from different clients are heterogeneous, considering factors such as variations in data distributions, label imbalance, and situations while some clients have limited training data available.
- Ablation Study: we conducted an ablation study to verify the advantages of employing a hypernetwork in MetaVD compared to naive Variational Dropout (VD) or Ensemble VD approaches in Federated Learning.
- Client Participation: we evaluated the performance of each algorithm under different levels of client participation rates in each FL round.
- Uncertainty Calibration: we used Expected Calibration Error (ECE) measures to evaluate the accuracy of probabilistic predictions made by the predictive models. These measures help determine whether a model is overconfident or underconfident in its predictions and identify any biases in the model's predictions.
- Communication Efficiency: we measured the communication overhead of each algorithm in terms of the rates of model compression.
- Multi-domain Datasets: Multi-domain learning aims to leverage all available training data across different domains to enhance the performance of the model, but it typically results in a suboptimal global model. We tested our approach in FL with multi-domain learning.
- Model Compression: we compared MetaVD's compression capabilities with the existing pruning algorithm and evaluated the performance of MetaVD with and without compression.

# G  Additional Results with Non-IID Settings

| Hetrogenity | $\dot{\alpha} = 5.0$ | | | $\dot{\alpha} = 0.5$ | | | $\dot{\alpha} = 0.1$ | | |
|---|---|---|---|---|---|---|---|---|---|
| Method | Test (%) | OOD (%) | $\Delta$ | Test (%) | OOD (%) | $\Delta$ | Test (%) | OOD (%) | $\Delta$ |
| FedAvg [1] | 42.35 | 43.08 | +0.73 | 41.92 | 41.96 | +0.04 | 37.57 | 37.90 | +0.33 |
| FedAvg+FT [18] | 41.49 | 42.45 | +0.96 | 40.99 | 39.83 | −1.16 | 36.79 | 37.00 | +0.21 |
| FedProx [73] | 42.23 | 44.11 | +1.88 | 42.03 | 40.51 | −1.52 | 38.07 | 39.36 | +1.29 |
| FedBE [31] | 45.17 | 45.43 | +0.26 | 44.29 | 44.23 | −0.06 | 40.89 | 41.56 | +0.67 |
| pFedGP [34] | 42.69 | 43.07 | +0.38 | 42.44 | 42.53 | +0.09 | 37.65 | 38.09 | +0.44 |
| Reptile [22] | 47.87 | 47.73 | −0.14 | 46.13 | 45.94 | −0.19 | 40.71 | 43.59 | +2.88 |
| MAML [21] | 48.30 | 49.14 | +0.84 | 46.33 | 46.65 | **+0.32** | 41.56 | 45.19 | +3.63 |
| PerFedAvg (HF-MAML) [23] | 48.19 | 47.35 | −0.84 | 46.22 | 46.36 | +0.14 | 42.46 | 42.92 | +0.46 |
| FedAvg+MetaVD (ours) | 47.82 | 50.26 | **+2.44** | 47.54 | 47.55 | +0.01 | 42.34 | 42.65 | +0.31 |
| Reptile+MetaVD (ours) | **53.71** | **54.50** | +0.79 | **52.06** | **51.50** | −0.56 | **44.56** | 44.62 | +0.06 |
| MAML+MetaVD (ours) | 52.40 | 51.78 | −0.62 | 50.21 | 49.75 | −0.46 | 43.84 | **48.17** | **+4.33** |
| PerFedAvg+MetaVD (ours) | 51.67 | 51.70 | +0.03 | 50.02 | 48.70 | −1.32 | 43.31 | 46.19 | +2.88 |

*CIFAR-100* dataset

Table 7: Results of classification accuracies with different (non-IID) heterogeneity strengths of $\dot{\alpha} = 5.0$, $\dot{\alpha} = 0.5$, and $\dot{\alpha} = 0.1$ in the CIFAR-100 dataset. We report the results of participating clients during the training (Test) dataset and non-participating clients (OOD). The higher, the better.

To evaluate the generalization capabilities of the models under non-IID data conditions, we conducted tests on both CIFAR-10 and CIFAR-100 datasets with varying degrees of heterogeneous partitions. We randomly selected 30 out of 130 clients as non-participating out-of-distribution (OOD) clients, who were not involved during the training phase. Table 11 and Table 12 display the weighted average accuracy while adjusting the non-IID degrees with different Dirichlet parameters $\dot{\alpha}$. A smaller $\dot{\alpha}$ corresponds to a higher degree of heterogeneity. Test(%) denotes the weighted average accuracy of participating clients' test samples, OOD(%) signifies the weighted average accuracy of the non-participants, and $\Delta$ represents the participation generalization gap calculated as the difference between OOD accuracy and test accuracy.

In both datasets, PFL methods such as Reptile, MAML, and PerFedAvg generally outperform non-PFL methods like FedAvg and FedProx. The Bayesian ensemble approach on FedAvg (FedBE) offers a slight improvement in overall performance but still lags behind PFL methods in terms of OOD accuracy. Notably, when combined with MetaVD, all baselines experience significant performance enhancements, regardless of whether they employ FL or PFL approaches. Importantly, our method consistently improves OOD accuracy across all degrees of non-IID data, illustrating its versatility in effectively augmenting conventional FL algorithms without being limited by specific optimization techniques, addressing non-IID data and model overfitting issues. Additionally, our experimental results indicate that pFedGP performs relatively poorly when the data distribution among clients is heterogeneous.

| Hetrogenity | *CIFAR-10* dataset | | | | | | | | |
|---|---|---|---|---|---|---|---|---|---|
| | $\dot\alpha = 5.0$ | | | $\dot\alpha = 0.5$ | | | $\dot\alpha = 0.1$ | | |
| Method | Test (%) | OOD (%) | $\Delta$ | Test (%) | OOD (%) | $\Delta$ | Test (%) | OOD (%) | $\Delta$ |
| FedAvg [1] | 80.73 | 79.98 | −0.75 | 78.28 | 78.90 | +0.62 | 71.65 | 71.57 | −0.08 |
| FedAvg+FT [18] | 80.20 | 79.63 | −0.57 | 76.96 | 75.83 | −1.13 | 69.62 | 68.38 | −1.24 |
| FedProx [73] | 80.79 | 80.30 | −0.49 | 78.55 | 77.60 | −0.95 | 72.27 | 73.75 | +1.48 |
| FedBE [31] | 81.55 | 81.07 | −0.48 | 79.44 | 79.46 | +0.02 | 70.23 | 69.19 | −1.04 |
| pFedGP [34] | 83.73 | 83.27 | −0.46 | 79.56 | 79.37 | −0.19 | 71.94 | 76.83 | +4.99 |
| Reptile [22] | 83.48 | 82.88 | −0.60 | 79.35 | 79.41 | +0.06 | 73.93 | 76.36 | +2.43 |
| MAML [21] | 82.61 | 81.69 | −0.92 | 80.31 | 80.03 | −0.28 | 76.06 | 74.89 | −1.17 |
| PerFedAvg (HF-MAML) [23] | 82.57 | 81.99 | −0.58 | 80.87 | 80.85 | −0.02 | 75.42 | 79.56 | +4.14 |
| FedAvg+MetaVD (ours) | 82.60 | 82.74 | +0.14 | 79.08 | 80.33 | **+1.25** | 76.87 | 76.25 | −0.62 |
| Reptile+MetaVD (ours) | **84.70** | 84.28 | −0.42 | **83.20** | **83.40** | +0.20 | 76.51 | **82.07** | +5.56 |
| MAML+MetaVD (ours) | 83.93 | **84.95** | +1.02 | 81.32 | 81.81 | +0.49 | **77.27** | 79.05 | +1.78 |
| PerFedAvg+MetaVD (ours) | 83.88 | 83.96 | +0.08 | 81.06 | 81.47 | +0.41 | 76.06 | 81.77 | **+5.71** |

Table 8: Results of classification accuracies with different (non-IID) heterogeneity strengths of $\dot\alpha = 5.0$, $\dot\alpha = 0.5$, and $\dot\alpha = 0.1$ in the CIFAR-10 dataset. We report the results of participating clients during the training (Test) dataset and non-participating clients (OOD). The higher the better.

In the original pFedGP paper [34], the data samples were partitioned across clients according to [25, 24], resulting in each client having 2 and 10 classes for the CIFAR-10 and CIFAR-100 datasets, respectively. In contrast, in our non-IID experiment, we do not predefine the number of classes each client has; instead, it is determined by the Dirichlet parameter $\dot\alpha$. In addition, we found some limitations of pFedGP, which could not classify data points when encountering unseen labels within a client (especially when the label shift [104, 105] occurs between the training and testing distributions of the clients). To ensure that pFedGP would run on the unseen labels, we circumvented this limitation by allowing the client to have access to 40 or 50 data points for each class while evaluating the model's performance.

| | Dirichlet parameter $\dot\alpha$ | | |
|---|---|---|---|
| Dataset | $\dot\alpha = 5.0$ | $\dot\alpha = 0.5$ | $\dot\alpha = 0.1$ |
| CIFAR-100 | $99.78 \pm 0.5$ | $73.18 \pm 5.2$ | $33.78 \pm 7.33$ |
| CIFAR-10 | $10 \pm 0$ | $9.10 \pm 0.94$ | $4.65 \pm 1.49$ |

Table 9: Statistics of the number of classes assigned to each client.

# H   Additional Ablation Results in FEMNIST Dataset

| | *FEMNIST* dataset | | |
|---|---|---|---|
| Method | Test (%) | OOD (%) | $\Delta$ |
| Reptile | 87.86 | 88.22 | **+0.36** |
| Reptile+VD | 87.93 | 85.88 | −2.05 |
| Reptile+EnsembleVD | 87.99 | 87.97 | −0.02 |
| Reptile+MetaVD (ours) | **89.43** | **88.71** | −0.72 |

Table 10: MetaVD ablation study results in the FEMNIST dataset.

In this experiment, we further performed the ablation study using the FEMNIST dataset to evaluate the benefits of MetaVD's hypernetwork in FL. We compared MetaVD to naive VD [46] and Ensemble VD [79] approaches. The naive (global) VD model keeps one global dropout parameter shared with all clients; the dropout parameter is treated as a global model parameter as in FedAvg. In EnsembleVD, we updated client-specific dropout rates in a manner analogous to the local adaptation step of MetaVD but maintained independent variational dropout rates for different clients on the server. In contrast, MetaVD utilized a hypernetwork to learn the dropout rates. All models employed Bayesian posterior aggregation rules based on dropout rates to update the global model parameter.

The ablation study's results on the FEMNIST dataset are outlined in Table 10 in the manuscript. It is evident from the table that MetaVD's conditional variational dropout-based hypernetwork surpasses all other baselines

in classification accuracy. Here too, Reptile+MetaVD consistently outperforms other methods. We noticed that in baselines like VD or EnsembleVD, client-specific dropout rates were not fully optimized due to restricted client participation. Consequently, the initial dropout rates and KL divergence loss in VD and EnsembleVD remained fairly static. Conversely, in MetaVD, both dropout rates and KL divergence loss converged in all tests. This observation underscores that MetaVD's hypernetwork presents a more data-efficient approach to learning the client-specific model uncertainty compared to other baselines.

# I  Additional Results on Uncertainty Calibration

Addressing uncertainty calibration issues is crucial for enhancing the accuracy and reliability of a model's predictions, particularly when the model is employed for making significant decisions. Uncertainty calibration metrics evaluate the accuracy of probabilistic predictions made by a predictive model. By identifying and addressing any calibration issues, it becomes possible to improve the model's prediction accuracy and reliability, which is of essential importance in decision-making processes. In a Federated Learning environment where clients have limited non-IID data, overfitting can easily occur, making the calibration of prediction uncertainty even more critical. We conducted experiments to evaluate the effectiveness of our proposed algorithm in improving uncertainty calibration, demonstrating its potential to strengthen model performance in various real applications.

| Hetrogenity | *CIFAR-100* dataset | | | | | |
| | $\dot\alpha = 5.0$ | | $\dot\alpha = 0.5$ | | $\dot\alpha = 0.1$ | |
| Method | ECE (%) | MCE (%) | ECE (%) | MCE (%) | ECE (%) | MCE (%) |
|---|---|---|---|---|---|---|
| FedAvg [1] | 0.29 | 22.05 | 0.53 | 32.35 | 0.60 | 36.79 |
| FedAvg+FT [18] | 0.41 | 28.36 | 0.46 | 31.91 | 0.69 | 45.04 |
| FedProx [73] | 0.43 | 21.92 | 0.40 | 26.74 | 0.67 | 39.69 |
| FedBE [31] | 0.43 | 29.75 | 0.38 | 27.54 | 0.50 | 34.66 |
| pFedGP [34] | **0.15** | 22.99 | **0.14** | **16.57** | **0.12** | **17.40** |
| Reptile [22] | 0.73 | 46.53 | 0.77 | 46.19 | 0.77 | 50.52 |
| MAML [21] | 0.40 | 27.74 | 0.70 | 48.16 | 0.75 | 46.47 |
| PerFedAvg (HF-MAML) [23] | 0.65 | 45.44 | 0.40 | 23.74 | 0.69 | 45.27 |
| FedAvg+MetaVD (ours) | 0.26 | **19.99** | 0.32 | 26.60 | 0.39 | 25.27 |
| Reptile+MetaVD (ours) | 0.37 | 25.24 | 0.32 | 23.86 | 0.57 | 42.40 |
| MAML+MetaVD (ours) | 0.31 | 23.18 | 0.38 | 32.56 | 0.52 | 37.26 |
| PerFedAvg+MetaVD (ours) | 0.26 | 20.46 | 0.39 | 27.59 | 0.43 | 30.20 |

Table 11: Results of uncertainty calibration scores (ECE and MCE) in the CIFAR-100 dataset. The lower is the better.

| Hetrogenity | *CIFAR-10* dataset | | | | | |
| | $\dot\alpha = 5.0$ | | $\dot\alpha = 0.5$ | | $\dot\alpha = 0.1$ | |
| Method | ECE (%) | MCE (%) | ECE (%) | MCE (%) | ECE (%) | MCE (%) |
|---|---|---|---|---|---|---|
| FedAvg [1] | 1.93 | 28.87 | 2.39 | 26.45 | 3.16 | 27.37 |
| FedAvg+FT [18] | 2.02 | 20.80 | 2.71 | 27.86 | 3.20 | 33.22 |
| FedProx [73] | 2.01 | 25.96 | 2.52 | 25.24 | 3.06 | 28.62 |
| FedBE [31] | 1.78 | 21.33 | 2.18 | 21.89 | 3.35 | 29.72 |
| pFedGP [34] | 1.23 | **8.34** | **0.91** | 6.23 | 0.84 | 6.08 |
| Reptile [22] | 2.40 | 30.53 | 3.06 | 38.89 | 2.75 | 27.76 |
| MAML [21] | 2.61 | 36.86 | 2.49 | 30.67 | 2.66 | 32.10 |
| PerFedAvg (HF-MAML) [23] | 2.19 | 26.78 | 2.39 | 27.13 | 2.58 | 24.47 |
| FedAvg+MetaVD (ours) | 1.52 | 18.02 | 1.76 | 21.17 | 1.40 | 13.92 |
| Reptile+MetaVD (ours) | 1.60 | 25.46 | 2.16 | 29.84 | 1.76 | 17.84 |
| MAML+MetaVD (ours) | 1.39 | 16.46 | 1.97 | 21.51 | 0.35 | 5.80 |
| PerFedAvg+MetaVD (ours) | **1.14** | 17.55 | 1.51 | **20.38** | **0.29** | **3.63** |

Table 12: Results of uncertainty calibration scores (ECE and MCE) in the CIFAR-10 dataset. The lower is the better.

Expected Calibration Error (ECE) and Maximum Calibration Error (MCE) are widely used metrics for measuring uncertainty calibration. ECE represents the average discrepancy between the model's confidence and its accuracy. To compute ECE, we group samples based on their confidence levels and calculate the average confidence and the percentage of correct samples for each group. ECE is then derived as the weighted average of the differences

between the average confidence and the percentage of correct samples across all groups. ECE values range from 0 to 1, where 0 signifies perfect calibration and 1 indicates complete miscalibration. MCE, on the other hand, is akin to ECE but focuses on the largest gap for any group rather than the weighted average.

Table 11 and Table 12 summarize the ECE and MCE results for varying non-IID degrees in CIFAR-100 and CIFAR-10 datasets, with Dirichlet parameters represented as $\dot{\alpha} = [5, 0.5, 0.1]$. Typically, ECE and MCE values tend to increase as clients possess more non-IID data, which is a result of overfitting each client's local data. In the CIFAR-10 dataset, all methods that incorporate MetaVD exhibit a decline in ECE and MCE values as $\dot{\alpha}$ decreases, in contrast to standard FL or PFL methods that show an increase in these values. MetaVD effectively mitigates overfitting to local data and enhances calibration by leveraging the heterogeneity present in the training data. Consequently, methods employing MetaVD consistently achieve the lowest ECE and MCE values in almost all scenarios.

Examining a reliability diagram can provide a visual comparison of uncertainty calibration results, as it illustrates the relationship between prediction probabilities and true labels. Ideally, if a model predicts a specific class with a certain probability, the actual label should correspond to that confidence level, placing the point on the reliability diagram's diagonal line. However, if the prediction probability and the true label do not align, the point would be below or above the diagonal line. In this context, ECE represents the average distance of each point from the diagonal line, while MCE indicates the maximum distance of any point from the diagonal line, offering a comprehensive understanding of the model's calibration performance.

Figure 4 and Figure 5 present the reliability diagrams for CIFAR-100 and CIFAR-10 with $\dot{\alpha} = 0.5$ respectively. These figures allow us to compare the calibration performance of FedAvg, Reptile, MAML, and PerFedAvg, both with and without the integration of MetaVD. Remarkably, in most instances, employing MetaVD leads to improved calibration, drawing the reliability diagrams closer to the identity function. This supports our findings that MetaVD significantly improves the uncertainty calibration of the models as well.

## J Additional Results on Muti-domain Datasets

| Hetrogenity | *CelebA + CIFAR-100* dataset | | | | | | | | |
| | $\dot{\alpha} = 5.0$ | | | $\dot{\alpha} = 0.5$ | | | $\dot{\alpha} = 0.1$ | | |
| Method | Test (%) | OOD (%) | $\Delta$ | Test (%) | OOD (%) | $\Delta$ | Test (%) | OOD (%) | $\Delta$ |
|---|---|---|---|---|---|---|---|---|---|
| FedAvg [1] | 43.42 | 44.45 | +1.03 | 43.65 | 43.45 | −0.20 | 40.46 | 41.00 | +0.54 |
| Reptile [22] | 50.40 | 49.07 | −1.33 | 48.92 | 48.93 | +0.01 | 43.41 | 42.16 | −1.25 |
| MAML [21] | 49.97 | 48.40 | −1.57 | 47.39 | 48.51 | +1.12 | 44.80 | 44.79 | −0.01 |
| PerFedAvg (HF-MAML) [23] | 50.61 | 49.61 | −1.00 | 49.21 | 50.91 | +1.70 | 44.20 | 44.13 | −0.07 |
| FedAvg+MetaVD (ours) | 48.08 | 48.88 | +0.80 | 48.23 | 48.62 | +0.39 | 42.98 | 45.19 | **+2.21** |
| Reptile+MetaVD (ours) | 53.04 | 53.97 | +0.93 | **52.26** | **54.75** | **+2.49** | 45.83 | 47.05 | +1.22 |
| MAML+MetaVD (ours) | 52.82 | 53.47 | +0.65 | 51.34 | 52.82 | +1.48 | **46.83** | **48.33** | +1.50 |
| PerFedAvg+MetaVD (ours) | **53.14** | **54.26** | **+1.12** | 51.18 | 53.06 | +1.88 | 46.65 | 47.11 | +0.46 |

Table 13: Classification accuracies with different (non-IID) heterogeneity degrees of $\dot{\alpha} = [5.0, 0.5, 0.1]$ in multi-domain datasets (a); CelebA + CIFAR-100.

| Hetrogenity | *CIFAR-100 + FEMNIST* dataset | | | | | | | | |
| | $\dot{\alpha} = 5.0$ | | | $\dot{\alpha} = 0.5$ | | | $\dot{\alpha} = 0.1$ | | |
| Method | Test (%) | OOD (%) | $\Delta$ | Test (%) | OOD (%) | $\Delta$ | Test (%) | OOD (%) | $\Delta$ |
|---|---|---|---|---|---|---|---|---|---|
| FedAvg [1] | 63.90 | 53.57 | −10.33 | 64.02 | 52.88 | −11.14 | 62.13 | 49.65 | −12.48 |
| Reptile [22] | 66.47 | 57.60 | −8.87 | 66.13 | 56.22 | −9.91 | 64.10 | 55.81 | −8.29 |
| MAML [21] | 66.86 | 53.25 | −13.61 | 66.56 | 55.72 | −10.84 | 64.56 | 56.21 | −8.35 |
| PerFedAvg (HF-MAML) [23] | 67.47 | 57.15 | −10.33 | 66.57 | 55.24 | −11.33 | 65.16 | 56.70 | −8.46 |
| FedAvg+MetaVD (ours) | 65.91 | 57.59 | −8.32 | 65.58 | 56.85 | −8.73 | 63.26 | 52.78 | −10.48 |
| Reptile+MetaVD (ours) | **68.61** | **60.95** | **−7.66** | **68.35** | 59.07 | −9.28 | 65.29 | 58.11 | **−7.18** |
| MAML+MetaVD (ours) | 68.41 | 58.86 | −9.55 | 68.24 | 61.21 | −7.03 | 65.46 | 58.07 | −7.39 |
| PerFedAvg+MetaVD (ours) | 68.27 | 58.72 | −9.55 | 67.93 | **61.32** | **−6.61** | **66.81** | **59.13** | −7.68 |

Table 14: Classification accuracies with different (non-IID) heterogeneity degrees of $\dot{\alpha} = [5.0, 0.5, 0.1]$ in multi-domain datasets (b); CIFAR-100 + FEMNIST.

Unlike existing federated learning algorithms that usually assume a single-domain approach where only one dataset is used in the experiment. In this section, we further evaluate the performance of our method on real-world non-IID FL experiments by introducing multi-domain datasets in which we assume each client can have data

| Hetrogenity | CelebA + CIFAR-100 + FEMNIST dataset | | | | | | | | |
|---|---|---|---|---|---|---|---|---|---|
| | $\dot\alpha = 5.0$ | | | $\dot\alpha = 0.5$ | | | $\dot\alpha = 0.1$ | | |
| Method | Test (%) | OOD (%) | Δ | Test (%) | OOD (%) | Δ | Test (%) | OOD (%) | Δ |
| FedAvg [1] | 63.57 | 53.10 | −10.47 | 63.73 | 55.55 | −8.18 | 61.86 | 52.98 | −8.88 |
| Reptile [22] | 65.93 | 57.98 | −7.95 | 66.98 | 57.12 | −9.86 | 63.89 | 52.33 | −11.56 |
| MAML [21] | 67.70 | 59.06 | −8.64 | 67.08 | 58.15 | −8.93 | 65.67 | 54.41 | −11.26 |
| PerFedAvg (HF-MAML) [23] | 67.39 | 59.17 | −8.22 | 67.20 | 57.03 | −10.17 | 65.08 | 54.54 | −10.54 |
| FedAvg+MetaVD (ours) | 63.62 | 55.87 | −7.75 | 65.93 | 58.58 | −7.35 | 64.59 | 55.61 | −8.98 |
| Reptile+MetaVD (ours) | 67.63 | 60.95 | −6.68 | 68.78 | 61.59 | −7.19 | 65.18 | 56.46 | **−8.72** |
| MAML+MetaVD (ours) | **69.42** | **61.52** | −7.90 | **68.81** | **61.60** | −7.21 | **66.56** | **56.54** | −10.02 |
| PerFedAvg+MetaVD (ours) | 67.83 | 60.74 | −7.09 | 68.05 | 61.17 | **−6.88** | 65.11 | 55.64 | −9.47 |

Table 15: Classification accuracies with different (non-IID) heterogeneity degrees of $\dot\alpha = [5.0, 0.5, 0.1]$ in multi-domain datasets (d); CelebA + CIFAR-100 + FEMNIST.

from different domains. Multi-domain learning [87–89] aims to leverage all available training data across different domains to enhance the performance of the model. However, directly utilizing data from different domains typically results in a suboptimal global model. In the context of federated learning, it becomes even more critical to apply multi-domain learning effectively. We use three different FL datasets to construct the multi-domain task distributions: CelebA, FEMNIST, and CIFAR-100. The non-IID heterogeneous environment has been also assumed in the field of multi-domain learning. We utilize the Dirichlet sampling technique ($\dot\alpha = [5.0, 0.5, 0.1]$) to sample each client's local CIFAR-100 data.

Table 13, 14, and 15 illustrate the classification accuracies for Test and OOD clients with varying degrees of heterogeneity. Employing MetaVD leads to significantly improved prediction accuracy, with larger improvements in OOD accuracy compared to the improvement in Test accuracy, across all multi-domain settings and degrees of heterogeneity. The results indicate that MetaVD can improve the versatility and performance of FL algorithms when applied to a broad range of multi-domain datasets.

# K    Additional Results on Model Compression

| Hetrogenity | CIFAR-10 dataset | | | | | | | | |
|---|---|---|---|---|---|---|---|---|---|
| | $\dot\alpha = 5.0$ | | | $\dot\alpha = 0.5$ | | | $\dot\alpha = 0.1$ | | |
| Method | Test (%) | OOD (%) | Sparsity (%) | Test (%) | OOD (%) | Sparsity (%) | Test (%) | OOD (%) | Sparsity (%) |
| FedAvg+SNIP | 81.03 | 80.46 | 70.0 | 78.53 | 79.30 | 70.0 | 71.67 | 73.13 | 70.0 |
| Reptile+SNIP | 83.21 | 82.10 | 70.0 | 81.43 | 81.72 | 70.0 | 74.56 | 69.57 | 70.0 |
| MAML+SNIP | 83.35 | 81.59 | 70.0 | 81.47 | 81.84 | 70.0 | 72.51 | 69.27 | 70.0 |
| PerFedAvg+SNIP | 83.33 | 83.49 | 70.0 | 80.97 | 82.47 | 70.0 | 76.27 | 75.90 | 70.0 |
| Reptile+MetaVD | **84.70** | 84.28 | 0 | **83.20** | 83.40 | 0 | 76.51 | 82.07 | 0 |
| MAML+MetaVD | 83.93 | 84.95 | 0 | 81.32 | 81.81 | 0 | 77.27 | 79.05 | 0 |
| PerFedAvg+MetaVD | 83.88 | 83.96 | 0 | 81.06 | 81.47 | 0 | 76.06 | 81.77 | 0 |
| Reptile+MetaVD+DP | 84.19 | 84.46 | 77.12 | 82.67 | 82.79 | 79.25 | 75.85 | 80.52 | 76.18 |
| MAML+MetaVD+DP | 84.37 | **84.95** | 76.76 | 82.65 | **83.64** | 78.85 | **77.77** | 80.74 | 72.47 |
| PerFedAvg+MetaVD+DP | 84.04 | 83.69 | 78.55 | 82.64 | 83.52 | 79.32 | 77.10 | **83.24** | 72.87 |

Table 16: Results of study on model compression with different (non-IID) heterogeneity strengths of $\dot\alpha = 5.0$, $\dot\alpha = 0.5$, and $\dot\alpha = 0.1$ in the CIFAR-10 dataset. MetaVD+DP does not communicate the model parameters when the dropout rate is larger than 0.9.

Federated Learning optimization involves frequent communication of model parameters between devices and the central server, which can be slow and may raise privacy concerns. Therefore, it is crucial to minimize communication costs by reducing both the size of exchanged model parameters and the number of communication rounds. MetaVD also has the benefit of compressing the model parameters needed for each client device. To explore the compression capabilities of MetaVD, we conducted experiments on CIFAR-10 and CIFAR-100 datasets, comparing our method to the baseline Pruning algorithm SNIP [101]. For the implementation of the SNIP algorithm in the context of Federated Learning, we referred to the work of Jiang et al. (2022) [102]; we let SNIP prune the original model to the target sparsity right after the first round.

Table 16 and Table 17 demonstrate the results of Test(%), OOD(%), and Sparsity(%) percentage of the models with or without MetaVD+DP. Sparsity represents the ratio of zero-valued model parameters in the personalized layer. The DP refers to the process of dropping communication of weights in the FL algorithm, where weights with a dropout rate greater than 0.9 are dropped after 150 rounds. We have set these 150 rounds to allow MetaVD

| Hetrogenity | $\dot{\alpha} = 5.0$ | | | CIFAR-100 dataset $\dot{\alpha} = 0.5$ | | | $\dot{\alpha} = 0.1$ | | |
|---|---|---|---|---|---|---|---|---|---|
| Method | Test (%) | OOD (%) | Sparsity (%) | Test (%) | OOD (%) | Sparsity (%) | Test (%) | OOD (%) | Sparsity (%) |
| FedAvg+SNIP | 43.76 | 44.11 | 70.0 | 42.53 | 41.54 | 70.0 | 37.80 | 40.34 | 70.0 |
| Reptile+SNIP | 49.54 | 49.61 | 70.0 | 48.38 | 48.62 | 70.0 | 42.03 | 43.68 | 70.0 |
| MAML+SNIP | 51.06 | 49.56 | 70.0 | 48.16 | 46.99 | 70.0 | 42.03 | 42.49 | 70.0 |
| PerFedAvg+SNIP | 50.66 | 48.85 | 70.0 | 48.71 | 48.46 | 70.0 | 43.23 | 43.09 | 70.0 |
| Reptile+MetaVD | **53.71** | 54.50 | 0 | **52.06** | **51.50** | 0 | 44.56 | 44.62 | 0 |
| MAML+MetaVD | 52.40 | 51.78 | 0 | 50.21 | 49.75 | 0 | 43.84 | **48.17** | 0 |
| PerFedAvg+MetaVD | 51.67 | 51.70 | 0 | 50.02 | 48.70 | 0 | 43.31 | 46.19 | 0 |
| Reptile+MetaVD+DP | 52.56 | **55.03** | 65.83 | 50.94 | 50.85 | 57.26 | 45.13 | 45.19 | 75.72 |
| MAML+MetaVD+DP | 52.31 | 52.29 | 74.61 | 50.30 | 50.88 | 77.88 | **45.75** | 45.78 | 72.26 |
| PerFedAvg+MetaVD+DP | 51.68 | 52.56 | 70.08 | 51.64 | 51.47 | 68.88 | 45.16 | 46.45 | 55.18 |

Table 17: Results of study on model compression with different (non-IID) heterogeneity strengths of $\dot{\alpha} = 5.0$, $\dot{\alpha} = 0.5$, and $\dot{\alpha} = 0.1$ in the CIFAR-100 dataset. MetaVD+DP does not communicate the model parameters when the dropout rate is larger than 0.9.

to adequately learn the client-specific dropout rates. The target sparsity of the SNIP algorithm was set to 0.7. Note that, in our MetaVD+DP setting, even though we dropped weights with a dropout rate greater than 0.9, the actual sparsity values are observed to be lower and tend to be around 70%, as indicated in the tables.

Overall, most models removed over 70% of their parameters in the personalized layer without losing much performance. In CIFAR-10, as heterogeneity increased, the performance gap between SNIP and MetaVD+DP tended to increase. For instance, in CIFAR-10 with $\dot{\alpha} = 5.0$, the OOD performance of SNIP model and MAML+MetaVD+DP model were $81.59$ and $84.95$, respectively, whereas in CIFAR-10 with $\dot{\alpha} = 0.1$, they were $69.27$ and $80.74$ respectively, demonstrating an increased difference in performance. This was similar in the case of Reptile+MetaVD+DP and PerFedAvg+MetaVD+DP. In CIFAR-100 dataset, the performance gap, according to the increase in heterogeneity, was not noticeable. Surprisingly, dropping model parameters could lead to performance improvements in some cases. For example, in CIFAR-100 with $\dot{\alpha} = 0.5$, the PerFedAvg+MetaVD+DP model's OOD performance increased from $48.70$ to $51.47$. This experiment demonstrates that the MetaVD method can decrease the communication cost in Federated Learning (FL) by compressing model parameters. This emphasizes the efficiency of MetaVD, this approach can be applied to numerous FL algorithms without significantly increasing communication costs. It also boosts the generalization performance for new, unseen clients. These aspects make our approach a significant addition to the field of Federated Learning.

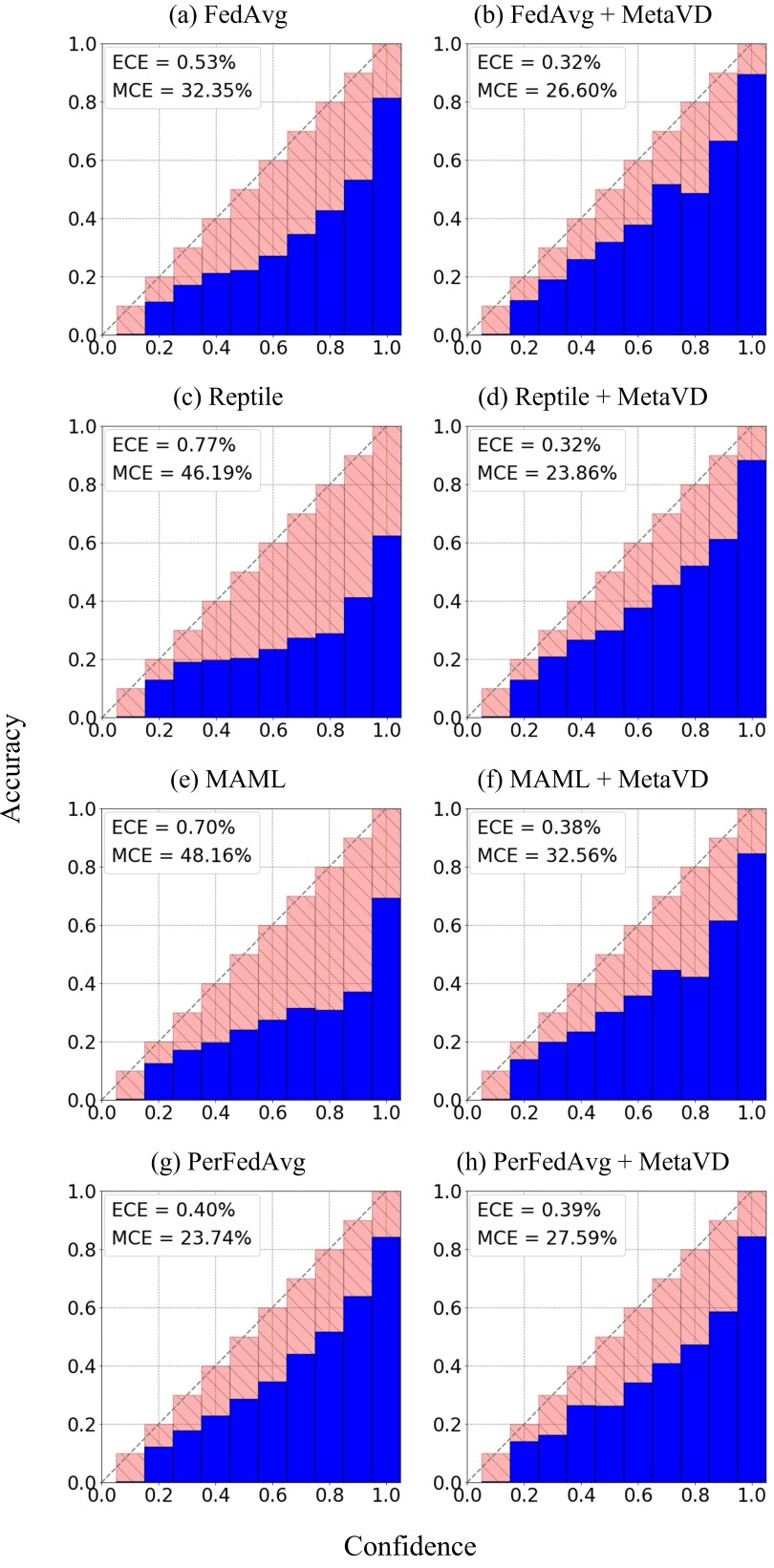

Figure 4: Reliability diagrams for (a) FedAvg, (b) FedAvg+MetaVD, (c) Reptile, (d) Reptile+MetaVD, (e) MAML, (f) MAML+MetaVD, (g) PerFedAvg and (h) PerFedAvg+MetaVD in CIFAR-100 ($\dot{\alpha} = 0.5$).

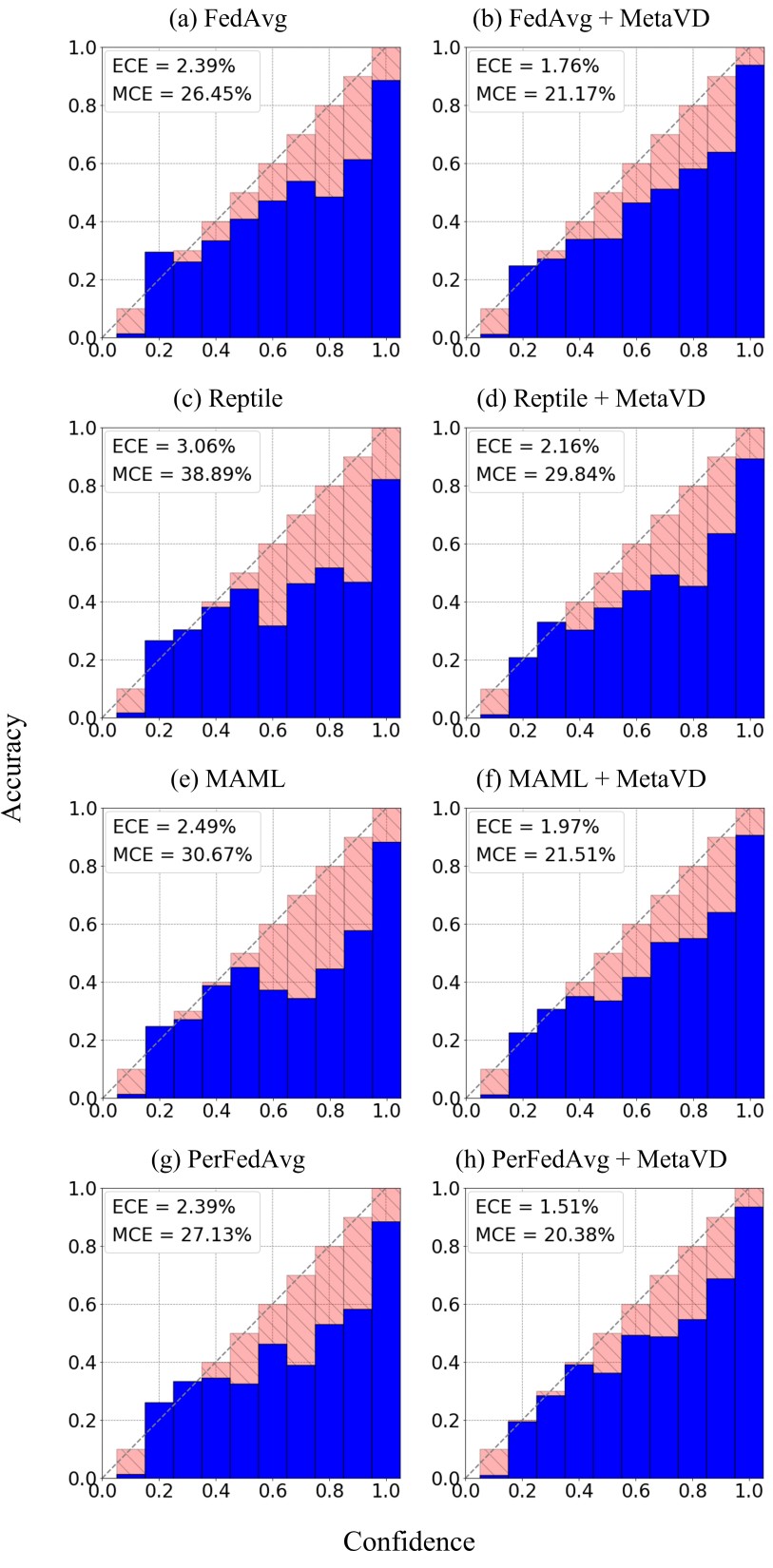

Figure 5: Reliability diagrams for (a) FedAvg, (b) FedAvg+MetaVD, (c) Reptile, (d) Reptile+MetaVD, (e) MAML, (f) MAML+MetaVD, (g) PerFedAvg and (h) PerFedAvg+MetaVD in CIFAR-10 ($\dot{\alpha} = 0.5$).

