# OpenReview forum: "Federated Learning via Meta-Variational Dropout"
_NeurIPS.cc/2023/Conference — NeurIPS 2023 poster_

### Official Review · Reviewer_mtpr · 2023-07-01

**Soundness:** 3 good
**Presentation:** 3 good
**Contribution:** 2 fair
**Rating:** 5
**Confidence:** 4

**Summary:**

The submission proposed a novel Bayesian meta-learning approach metaVD for federated learning. metaVD learns to predict client-dependent dropout rates via a hypernetwork, helping address the model personalization and limited non-i.i.d. data problems. At the same time, metaVD compressed the model parameter, alleviated the overfitting and reduced the communication costs.

**Strengths:**

MetaVD encompasses a new posterior aggregation strategy to consolidate local models into a global one. In addition, MetaVD predicts the dropout rates of parameters via a hypernetwork, enabling parameter compression. This not only allievates the overfitting problem but also reduces the communication costs of exchanging model parameters.

**Weaknesses:**

In general, this article is written in a fluent, simple, and easily understandable manner. However, there are two main issues that need to be addressed:

1. The proposed method in the article is straightforward and intuitive, but many aspects lack theoretical guarantees and analysis.

2. In the past years, Bayesian federated learning has made significant progress, but many relevant works have not been discussed or compared in the article.

3. if my understanding is right, the propsoed method has the risk of data leakage.

**Questions:**

1. “Recently, the Bayesian learning paradigm was introduced to the FL to tackle overfitting by considering the uncertainty of the model parameters [28–30]. However, they could also struggle with diverging local models if the data from different clients exhibit significant statistical variability.” This statement is incorrect. In recent years, Bayesian federated learning has made significant advancements. Many Bayesian methods have been employed not only to address overfitting and introduce uncertainty but also to tackle personalized scenarios. I list some works as reference below.

2. "Moreover, previous Bayesian FL methods are not developed for model personalization, and the performance can be degraded in non-i.i.d. client data." This statement is incorrect for the above reasons.

3. Why was the variational distribution in Equation 3 designed in such a form? I understand that this is an engineering approach, but can this parameterized form effectively approximate the posterior distribution of the weights? Is there any theoretical analysis available? Furthermore, why is the variance parameter $\alpha_m$ used to model the dropout rate? Is there any inherent connection between the two? While this may be an engineering approach, is there any theoretical or intuitive analysis supporting it? It is important to note that while engineering approaches are often driven by empirical performance, they can also be guided by theoretical insights or intuitions.

4. In eq(5), what is the definition of $g_m$? How does the expression of $r_k^m$ come out? Any derivation?

5. In the server-side optimization, when you update the parameters $\psi,e^m$, you are trying to optimize the $L_{ELBO}^m$ w.r.t. $\psi$ and $e^m$, but $L_{ELBO}^m$ requires the data on each client, doesn't this imply data leakage?

6. In the experiment section, as I mentioned earlier, this article lacks thorough investigation into Bayesian federated learning (FL), as it fails to mention and compare many Bayesian FL approaches specifically designed to address personalized scenarios. To name a few examples but not exaustive:
[1] Personalized federated learning with gaussian processes, NIPS 2021.
[2] Federated Bayesian Neural Regression: A Scalable Global Federated Gaussian Process. H Yu et al. 2022.
[3] Personalized federated learning via variational bayesian inference, ICML, 2022.

**Limitations:**

The authors discussed the limitations of their work, but they did not address the potential negative societal impact it may have.

---

> ### Author Rebuttal · Authors · 2023-08-10
>
> **We thank the reviewer for insightful comments.**
>
> Here are answers to your questions:
>
> **1. Bayesian PFL work is not discussed.**
>
> When we started our research last year, we were aware of only a few papers [17,15] dealing with the personalization of models using Bayesian methods. Meanwhile, we realized that measuring the algorithm’s performance on unseen clients during training is an important task [5]. In real FL applications, most clients may not participate in training. Then pFL benchmark paper [1] also extensively analyzed the gap between participating and non-participating clients across various pFL scenarios. Thus, we adopted their pFL evaluation protocols for our experiment.
>
> On the other hand, pFedBayes [15] did not show results for OOD clients. They assume an independent local model for each device while learning a shared global model. In most of their experiments, the performance of local models showed better results than the global model, but there are no local models for unseen clients. pFedGP [17] learns a shared kernel for all clients and infers personalized classifiers for each client. However, the generalization gap in Figure 4 seems large (compared to our results in Table 2). Also, the inducing point (IP) approximation to speed up the computation of inverting a large kernel matrix did not show better results than the original pFedGP [17] in their OOD experiment.
>
> For this reason, we focused on the meta-learning-based PFL methodology for most of our baseline because this approach is a researched methodology to apply to out-of-distribution (OOD) tasks. Also, we adopted the product of the posterior strategy for the global model aggregation. The posterior aggregation approach considers the uncertainty of weight in the model aggregation. However, pFedBayes and pFedGP update the global parameter (or the kernel) similar to FedAVG. That is why we cited Bayesian FL works related to posterior aggregation at most.
>
> We had no intention of disregarding the existing Bayesian PFL works. We will add a thoughtful discussion paragraph about the Bayesian PFL works and report the results of pFedGP [17] as a baseline (we could not find the official codes for the other papers). Only one article [19] in the arXiv addresses PFL with the Bayesian meta-learning (updated in July 2023 and not yet officially published). Please note that our work is still novel in this direction as our work is the first approach to utilize the variational dropout uncertainty in the model aggregation in the Bayesian PFL.
>
> **2. The introduction about Bayesian FL.**
>
> The “previous Bayeain FL” means those approaches mentioned in the same paragraph. We will change the sentence or add an explanation accordingly. (e.g., “Recently, Bayesian PFL methods [17,15,18] are introduced in this field and have been shown effective in dealing with non-IID. However, their approach still has limitations (e.g., Strong constraints in pFedBayes and high computational cost in pFedGP). Their model aggregation rule does not specifically considers the model uncertainty.” )
>
> **3. The variational distribution in equation 3.**
>
> As in L123, we extended the variational dropout (VD) [10, 21, 12, 22] approach for the local posterior. The VD is a regularization technique in Neural Networks (NN). This involves randomly turning off NN parameters during training by applying discrete Bernoulli random noises; a method initially used to prevent over-fitting [C].
>
> Later, it was found that continuous noises sampled from Gaussian distributions work similarly to the Bernoulli dropout due to the central limit theorem [A, B]. The $m$-th client's weight is interpreted as a form of Gaussian posterior distribution $q(w^m) = \mathcal{N}(w^{m} \vert \theta, \alpha^{m} \theta^2)$ where a mean equal to the global NN's weight $\theta$, and variance is the square of the weight (i.e., $\theta^2$) times the client-specific dropout variable $\alpha^m$.
>
> [A] Fast dropout training. ICML, 2013.
>
> [B] Regularization of neural networks using dropconnect. ICML, 2013.
>
> [C] Dropout: a simple way to prevent neural networks from overfitting. The journal of machine learning research, 2014.
>
> **4. The derivation of the aggregation rule in equation 5.**
>
> The definition of $g^m$ is a weighting factor of each client proportional to the local data size (mentioned in L56 and L108).
> We presented the derivation of the aggregation rule in the general comment.
>
> **5. The risk of data leakage?**
>
> In the FL system, the server and client only exchange the parameters (summarized statistics) instead of raw data, thus reducing the risk of data leakage. In our study, MetaVD only exchanges the NN parameter $\theta$ and dropout parameter $\alpha^m$. In addition, the dropout is applied to the weight of just one layer before the output layer (L202). Therefore, the risk of data leakage is relatively small or negligible compared to other existing FL baselines.
>
> Computing the gradient w.r.t $\psi$ and $e^m$ happens in the server after the server receives $\theta^m_*$ and $\alpha^m_*$. The expression in L171 $\nabla_\psi \mathcal{L}^m_{\text{ELBO}}(\alpha^m) \approx (\nabla_\psi \alpha^m)^{T} \Delta \alpha^m$ is an approximation of computing gradient w.r.t $\psi$. $\nabla_\psi \alpha^m$ is the gradient of a hyper network's output w.r.t $\psi$.  $\Delta \alpha^m = \alpha_*^m - \alpha^m$ is the changes in the local dropout variable, approximating the vector-Jacobian product. For this updating rule, we followed the recently proposed PFL algorithm, pFedHN [5] (please see Section 3.2 of them). In this work, we extended the hyper-network to approximate the dropout variable (or conditional variance) to fully utilize the weight uncertainty in the (aggregation, regularization, and personalization of) FL.
>
> **6. The investigation into Bayesian PFL.**
>
> (Please see "1" above and the Bayesian PFL in the general comment)
>
> **Reference**
>
> (Please use the reference list in the general comment)

---

### Official Review · Reviewer_yoXe · 2023-07-04

**Soundness:** 4 excellent
**Presentation:** 4 excellent
**Contribution:** 2 fair
**Rating:** 6
**Confidence:** 4

**Summary:**

This paper proposes a new federated learning (FL) framework to address the various issues of FL. Specifically, this framework (a) leverages Bayesian FL to address the issue of non i.i.d. data among clients, (b) instantiates its BFL model with a variational dropout posterior to efficiently handle large amount of clients, and (c) applies meta learning adaptation strategies (e.g., MAML and Reptile) at client-sides to personalize the local models. The paper demonstrates the performance of this approach on several vision datasets, with ablation studies to confirm the effectiveness of proposed components.

**Strengths:**

The proposed method makes sense and addresses various relevant problems in the field of FL. I think the paper did an excellent job combining various ideas, and presenting them as one coherent framework. In addition, the empirical results positively improve over existing baselines, which suggests strong practical merits. I also like that the paper is really rigorous with conducting ablation studies.

**Weaknesses:**

The most apparent weakness of this paper is the lack of technical novelty. Bayesian FL, Variational Dropout, and Meta-learning are all very well-known strategies. While I am not opposed to a creative combination of ideas, such idea should have a scientific merit that outweighs the sum of its individual components. Here, each component is behaving exactly as expected and there seems to be no technical challenge in terms of integrating them. I think the novel adjustment here is the hypernetwork (although I am not sure if this strategy has been previously adopted in the context of variational dropout --- e.g., hierarchical prior is sort of similar, but not exactly the same).

Some other weaknesses:
- Lack of error bar/standard deviation in any of the reported results.
- I would prefer some sort of performance vs. communication rounds to demonstrate convergence. Result tables are acceptable, but very uninformative.
- It was previously claimed that the hypernetwork helps with large number of clients, especially when some of them have limited data. I think this should be supported by an ablation study showing the effect of increasing no. clients/ decreasing client data (the current ablation study only compares MetaVD to NaiveVD/EnsembleVD on the default client setting).

Some other minor issues:
- Various typos (e.g., FEMINIST, line 187; Hetrogenity, table 2 header ...)

**Questions:**

- What is the benefit of the hyper-network, as opposed to directly learning the client-specific dropout rates ?
- What is the overhead computation cost of the proposed framework? I believe both local meta-learning and training a server-side hypernetwork is quite costly, and it is quite unfair to compare 1000 comm rounds of MetaVD to 1000 comm rounds of FedAvg.

**Limitations:**

The authors have discussed some limitations of the work in the conclusion. I think the method is purely theoretical and there is no potential negative societal impact.

---

> ### Author Rebuttal · Authors · 2023-08-10
>
> **Thank you for the helpful review!**
>
> In this work, we extended the hypernetwork to approximate the dropout variable of NN's weight to enable the full utilization of the weight uncertainty in the aggregation, regularization, and personalization of FL. Our efficient composition is also implemented on various meta-learning based baselines and PFL benchmark datasets.
> Here we briefly summarize the contribution of our works.
>
> **Contribution**
> * In FL, we show a new hyper-network-based Bayesian network approach is feasible.
> * Our method is the first approach of employing the variational dropout uncertainty in the posterior aggregation for the PFL.
> * We show that the conditional Gaussian (dropout) noise injection to NN weight method is a versatile approach that can be combined with any optimization-based meta-learning approach.
> * Experimentally, we have verified the performance of our work in various PFL benches. We release the experiment code for all baselines for reproducibility. We put much effort into implementing the stable gradient propagation to hypernetwork and the global parameter of meta-learning algorithm using the PyTorch functorch library.
>
> **Lack of error bar/standard deviation.**
> We presented the results of baselines with the best parameter settings optimized via a hyperparameter search tool, Optuna. We performed an extensive analysis of various different FL scenarios.
>
> **An ablation study showing the effect of increasing no. clients/ decreasing client data.**
> Some portions of clients have small datasets in the non-IID setting. They increase with a higher alpha (Dirichlet parameter). The reason we reported the simple ablation study is that the EnsembleVD cannot actually learn the independent dropout parameters well. So, technically, EnsembleVD was reduced to FedAVG with Dropout(0.1).
>
> **1. The benefit of the hypernetwork.**
>
> In our experiment, the dropout rates in EnsembleVD were not well optimized due to restricted client participation. On the other hand, joint optimization of MetaVD with hypernetwork converges well. This can be observed via the changes in the KL divergence loss term.  This demonstrates that MetaVD's hypernetwork offers a more data-efficient approach to learning the client-specific model uncertainty than Ensemble VD.
>
> **2. What is the overhead computation cost of the proposed framework?**
>
> We found that all models converged in 1000 rounds in our experiment. The only difference between MetaVD and FedAVG is one layer before the last layer. Since we only applied the conditional dropout to only the last layer, the increase in computation cost is quite small.
> The actual (approximate) training time in Cifar-10 dataset is  4 hours for  FedAVG, 5 hours for MetaVD, 6 hours for PerFedAvg (pFedGP in our experiment seems to take almost 12 hours).

---

### Official Review · Reviewer_1ob4 · 2023-07-06

**Soundness:** 3 good
**Presentation:** 3 good
**Contribution:** 3 good
**Rating:** 6
**Confidence:** 5

**Summary:**

This paper proposes a novel Bayesian personalized federated learning approach using meta-variational dropout. The proposed approach employs a shared hypernetwork to predict the client-dependent dropout rates for each model parameter, enabling effective model personalization and adaptation in the limited non-i.i.d. data environment. The effectiveness of this approach is demonstrated empirically on various FL datasets, including CIFAR-10, CIFAR-100, FEMINIST, and CelebA and multi-domain FL datasets.

**Strengths:**

1. The paper is clearly written and well organized.

2. The proposed method in the paper is well-motivated and technically correct.

3. The Bayesian approach to FL is interesting and seems suitable for personalization. The introduced MetaVD seems to work well in practice.

4. The method is extensively tested on a variety of FL datasets.



**Weaknesses:**

1. The discussion with related work needs to include other Bayesian treatments for personalized FL, e.g. [1], [2]

2. The experiments lack a comparison with relevant methods mentioned above [1-2], as well as the baseline in [3].

[1] Personalized Federated Learning via Variational Bayesian Inference. ICML 2022.

[2] Fedpop: A Bayesian approach for Personalised Federated Learning. NeurIPS 2022.

[3] Personalized Federated Learning using Hypernetworks. ICML 2021.

**Questions:**

1. Page 3, line 115: i.e., $\phi=\phi^1,..,\phi^M$ should be $\phi=\{\phi^1,..,\phi^M\}$.

2. Page 3, line 132: add a definition of $\mathbf{1}$ for  $\epsilon^m \sim \mathcal{N}(\mathbf{1}, \alpha^m)$ .
3. In equation (5), how was the aggregation rule derived? It is recommended to provide more details (referring to [4] is possible).

4. It would be clearer to specify the specific sections of the Appendix. For example, on page 6, "More details are in Appendix."
5. The sparsity in Table 7 seems to only consider the personalized layer (the last fully connected layer). I am curious about the proportion of discarded parameters to the total parameters of the model.
6. I hope the authors can provide a discussion on the communication complexity of the additional hypernetwork.
7. Algorithm 1 seems confusing. It would be clearer to separate Algorithm 1 into two distinct algorithms.



[4] A Bayesian Federated Learning Framework with Online Laplace Approximation.

**Limitations:**

NA.

---

> ### Author Rebuttal · Authors · 2023-08-10
>
> **We thank the reviewer for the positive and encouraging feedback!**
>
> As the review suggested, the recommended recent (Bayesian) PFL works will be thoroughly compared in the Background section.
> We will update the difference between the recent PFL approaches [1,2,3,4] and ours.
> We will also add the experimental results of pFedGP [17] as a Bayesian PFL baseline, as we could not find officially released codes for the other methods.
>
> Here are answers to your questions:
>
> **1. The correction for L115.**
>
> Thank you for the correction.
>
> **2. A definition of for  $\epsilon \sim N(\mathbf{1}, \alpha)$ in L132.**
>
> $\mathbf{1}$ is a K dimensional all-ones vector. K is the dimension of the NN's weight. ($\vec{\mathbf{1}}$ might be appropreate.)
>
> **3. The derivation of the aggregation rule.**
>
> Thank you for the suggestion! Please see "the derivation of the aggregation rule" in the general comment.
>
> **4. Specify the specific sections of the Appendix.**
>
> We will add the section number for all the references to Appendix.
>
> **5. The sparsity in Table 7.**
>
> Yes, it is applied to the personalized layer before the last layer. Here is the computation of the proportion: 12900 (params of a personal FC layer) * 0.8 (dropout rate) / 416612 (total params) * 100 = 2.48 (\%) approximately. We are showing the possibility of not increasing the model size while incorporating the additional model uncertainty parameter in Bayesian PFL.
>
> **6. Communication complexity of the additional hypernetwork.**
>
> The server only sends the global parameter and the approximated dropout variable to the client while keeping the hyperparameter in the server. Hence, if we apply dropout to the full NN layers, the communication complexity of our work is $\mathcal{O}(2K)$ where K is the dimension of full NN parameters. This doubles the cost compared to FedAVG. This is due to the modeling of uncertainty in the Bayesian framework similar to predates. However, our MetaVD is applied to just a single layer before the last output and also can prune some of the parameters. The increase in communication costs could be negligible.
>
> **7. Algorithm 1 seems confusing. It would be clearer to separate Algorithm 1 into two distinct algorithms.**
>
> Thank you for the suggestion. We will try to optimize the readability of Algorithm 1.
>
> **Reference**
>
> (Please use the reference in the general comment)

---

> > ### Comment · Reviewer_1ob4 · 2023-08-20
> >
> > Thank you for your response. Considering other reviewers have also mentioned concerns regarding the readability and details, I will maintain the current score.

---

### Official Review · Reviewer_6ZgF · 2023-07-06

**Soundness:** 3 good
**Presentation:** 4 excellent
**Contribution:** 2 fair
**Rating:** 5
**Confidence:** 4

**Summary:**

The paper introduces a novel approach called meta-variational dropout (MetaVD) for addressing challenges in federated learning (FL). Traditional FL faces issues such as model overfitting and divergence of local models due to non-i.i.d. data across clients. MetaVD leverages Bayesian meta-learning to predict client-specific dropout rates using a hyper network. Extensive experiments conducted on various FL datasets demonstrate that MetaVD achieves good performance.

**Strengths:**

1. The proposed approach is novel.
2. Authors performed experiments on a variety of tasks and datasets and showed promising results.
3. The proposed method showed consistent improvements over the considered baselines with a good margin.

**Weaknesses:**

A primary weakness is that the authors have not compared their results with existing state-of-the-art personalized FL algorithms, e.g., [1,2] and the related work/baselines referenced within them.

Other notes and requests for clarification:
1. In L66, where the authors discuss the convergence guarantees of the original FL algorithm, it is important to provide a citation.
2. The authors have written about the “Challenges of FL” in Section 2. They should briefly describe how the proposed approach addresses these challenges.
3. Figure 1 requires additional details to aid understanding. The authors should clarify the meaning of the "x" in the modules of the figure and explain how it illustrates the difference in aggregation between MetaVD and FedAvg.
4. In Table 3, the signs of "+" and "-" should be reversed to ensure accurate representation.
5. Table 4 should explicitly mention that the results presented are for out-of-distribution (OOD) clients to provide a clear context for the findings.
6. Figure 1 is not referenced or discussed in the paper. It should be either referred to in the main text or removed if it does not contribute significantly to the paper's content.
7. Additional information regarding the computation of gradients for the hypernetwork parameters should be included.
8. The authors should provide an explanation as to why the PerFedAvg+MetaVD+DP combination in Table 7 leads to improved performance despite dropping almost 80% of the parameters.
9. Regarding equation 5, the authors should elaborate on the implications of the inverse dependence of the aggregation weights on the square of the model weight. This information would enhance the understanding of the aggregation process and its impact on the final results.


[1] “Fusion of Global and Local Knowledge for Personalized Federated Learning”, TMLR 2023.

[2] “FedALA: Adaptive Local Aggregation for Personalized Federated Learning”.

**Questions:**

(see "weaknesses" above).

**Limitations:**

The authors haven’t discussed the limitations of their work.

---

> ### Author Rebuttal · Authors · 2023-08-10
>
> Thank you for the encouraging feedback on our work!
>
> We appreciate introducing the important PFL works (FedSLR, FedALA). We will cite them in the paper.
>
> **PFL Baseline**
> We summarize the baselines and related works **presented** in the submitted paper
> |**Method**|category|status|
> |-----------|--------|------|
> |FedAvg|FL|baseline|
> |FedProx|FL|baseline|
> |FedAvg+FT [1]|Personalized FL|baseline|
> |(pFed) Reptile [2]|Personalized FL|baseline|
> |(pFed) MAML [3]|Personalized FL|baseline|
> |PerFedAvg [4]|Personalized FL|baseline|
> |pFedHN [5]|Personalized FL|cite|
> |FedAG |Bayesian FL|cite|
> |FedBE |Bayesian FL|baseline|
> |FedPA [11]|Bayesian FL|cite|
> |FedAVG + SNIP [9]|FL + Pruning|baseline (in appendix)|
> |MetaVD (ours)|Bayesian PFL|ours|
>
> * Reptile [2], MAML [3], and PerFedAvg [4] are the SOTA PFL baselines extended from few-shot/meta-learning approaches. We chose these meta-learning-based PFL approaches as our key baseline since they perform well on out-of-distribution (OOD) clients. A summary of baselines is presented in section D of the appendix.
>
> * We extensively analyzed all baselines to find the best parameter settings using a hyperparameter search tool, Optuna. We conducted experiments on various FL algorithm evaluation scenarios following the recent FL benchmark paper, pFL-Bench, which makes our approach comparable to 20 existing SOTA pFL methods experimented in the pFL-Bench paper. For example, our results in Table 4 on FEMNIST are comparable to Table 2 of the pFL-Bench paper. In the CIFAR-10 and CIFAR-100 experiments, the sample sizes for participating and non-participating clients are slightly different; otherwise, they are almost the same.
>
> Here are answers to your questions:
>
> **1. L66.**
>
> In L22, we cited two references; one discusses the convergence rate of FedAvg with IID data, and the other is a survey paper.
>
> **2. “Challenges of FL” in Section 2.**
>
> * Heterogeneity of client data: MetaVD is a technique of modulating a global NN parameter (of FedAvg, or (pFed) Reptile/MAML) with a conditional dropout technique. Predicting the client-specific dropout variable via a hypernetwork enables reconfiguring a single NN for various different tasks. We also utilized the client-specific dropout rate as a weighting factor in the global model aggregation.
>
> * Sparse connectivity (or client’s participation in training): Table 5 in our experiment illustrates that reducing the participating client size does not degrade the prediction scores when we applied MetaVD. We hypothesize this is partially due to the conditional dropout preventing the overfitting into a small subset of clients.
>
> * limited data: The initial parameters learned in meta-learning help predict a local model with only a few data adaptations; thus, meta-learning-based PFL is advantageous to the client with limited data. The conditional dropout also prevents the local model overfitting by additionally regularizing (or conditioning) the initial parameters.
>
> * Communication cost: We demonstrated the possibility of pruning the model parameter in FL communication.
>
> **3. Figure 1 requires additional details.**
>
> We will update Figure 1 as follow:
>
> * We will add the meaning of “X” in Figure 1: it represents the pruned parameters in each communication round.
> * When aggregating local parameters, MetaVD utilizes the model’s weights as well as the weights’ uncertainty to find its global optimum. This corresponds to the global posterior mean of the product of two local Gaussian posteriors. In contrast, FedAVG only considers the mean of each local model. We will add this description with more details to the figure to explain the difference in the global aggregation between them.
>
> **4. The reversed sign in Table 3.**
>
> Thank you! We will reverse the sign.
>
> **5. Specify OOD clients in Table 4.**
>
> We will update the phrase (L261) in the caption.
>
> **6. Reference Figure 1.**
>
> (see “3.” above)
>
> **7. The computation of gradients for the hypernetwork parameters.**
>
> (Please see ``the update rule for hypernetwork'' in the general comment)
>
> **8. Why the PerFedAvg+MetaVD+DP combination in Table 7 leads to improved performance.**
>
> The performance enhancement after pruning our model aligns with Occam's Razor, which favors simpler explanations and helps prevent overfitting. This principle explains why pruning or dropout training can increase performance, a phenomenon also observed in existing works [9, 13]. A comparison with the SNIP pruning algorithm [9] is included in the Appendix on page 9.
>
> **9. The implications of the inverse of $\theta^2$ in Eq 5.**
>
> In our work, the $m$-th client's posterior distribution is defined as Gaussian dropout posterior [10,12] of the form $q(w^m) = \mathcal{N}(w^{m} \vert \theta, \alpha^{m} \theta^2)$ where a mean equal to the global NN's weight $\theta$, and variance is the square of the weight (i.e., $\theta^2$) times the client-specific dropout variable $\alpha^m$. Thus, the $\theta^2$ emphasizes the effect of larger weights on the estimated variance $\sigma^2 = (\alpha^m)\theta^2$.
>
> The $\alpha^m(\theta^m)^2$ in Eq 5 describes the estimated uncertainty in the model parameter for the $m$-th client. The inverse of this term is used to compute the aggregation weights, $r^m$. This means that parameters with more uncertainty (larger values or higher dropout rates) will have less influence on the aggregated global model parameter $\theta^{\text{agg}}$. Based on this technique, the federated learning system can adaptively adjust each client's local model's contribution to the global model. This method can help reduce the impact of outliers or anomalous clients that may have deviating model parameters due to unique local data distributions or noise. Also, in statistical learning theory, the Fisher information is directly proportional to the precision (or inverse variance) of Gaussian distribution.
>
> **Reference**
> (Please use the reference in the general comment)

---

### Author Rebuttal · Authors · 2023-08-10

# General comment to all reviewers

We sincerely thank the reviewer for our work's positive and encouraging feedback.

Here, we summarize some common answers to the reviewers.

## Bayesian PFL
Since we hear the reviewers' suggestion of discussing some recent Bayesian PFL works, we will add a thoughtful discussion paragraph about the recent development in Bayesian PFL works.
We will remove Table 1 in the Background section and add a paragraph to distinguish our work.
We will report the results of pFedGP[17] as a baseline in the experiment (we could not find the official codes for the others).
We can find only one paper [19] in the arXiv addressing PFL with the Bayesian meta-learning (updated in July 2023 and not officially published). Our work is still rare in this direction. Also, our work is the first approach to utilize the conditional variational dropout uncertainty in the model aggregation in the Bayesian PFL.

|**Method**|Category|Status|
|-----------|-----------|-------|
|pFedGP [17]|Bayesian PFL|→ baseline|
|pFedBayes [15]|Bayesian PFL|→ cite|
|FedBNR [18]|Bayesian PFL|→ cite|
|Fedpop [16]|Bayesian PFL|→ cite|
|FedPPD |Bayesian PFL|→ cite|
|FedABML [19]|Bayesian Meta PFL|→ cite|
|MetaVD (ours)|Bayesian Meta PFL|ours|

## The aggregation rule in Eq 5

Eq 5 is derived from the product of Gaussian local posteriors.

 In general, the mode of the product of $M$ Gaussians, $\prod_{m=1}^{M} \mathcal{N}(\mu_m, \sigma_m)$, simplifies to $\mu_{\text{agg}} = \sum_{m=1}^M r_m \mu_m$  where $r_m = (\sigma^2_m)^{-1} / \sum_{m=1}^M (\sigma^2_m)^{-1}$.

It is a popular statistical theory [23] (mentioned in L160).

The difference in our approach is that we utilize the Gaussian dropout posterior,  $q(w^m) = \mathcal{N}(w^{m} \vert \theta^m, \alpha^{m} (\theta^m)^2)$, for each m-th client, where $\theta^m$ and $\alpha^m$ are locally updated NN's weights and dropout parameters, respectively. If we consider the mean $\mu_m = \theta^m$ and variance $\sigma_m^2 = \alpha^m (\theta^m)^2$, we can get Eq 5 as an approximation of the mean of aggregated posterior.

In Eq 5, the definition of  $g^m$ is a weighting factor of each client proportional to the local data size (mentioned in L56 and L108). This formulation is directly adapted from Eq 3 of the paper FedPA [11] and [6,7]. In fact, minimizing the local posteriors using the KL divergence in L112 is equivalent to maximizing the logarithm of the product of weighted posteriors as described in Eq 8 of [7]. In the heterogeneous data, the product rule can achieve a smaller aggregation than the mixture of Gaussian posteriors. We can also find a similar derivation in the continual learning paper [8].

## The update rule for hypernetwork
The expression in L171 is an approximation of computing gradient w.r.t $\psi$.: $\nabla_\psi \mathcal{L}^m_{\text{ELBO}}(\alpha^m) \approx (\nabla_\psi \alpha^m)^{T} \Delta \alpha^m $ where $\nabla_\psi \alpha^m$ is the gradient of a hyper network's output w.r.t $\psi$ and $\Delta \alpha^m = \alpha_*^m - \alpha^m$ is the changes in the dropout variable, where  $\alpha_*^m$ is the locally updated dropout rate using the local data and  $\alpha^m$ is the initially predicted dropout variable from hyper-network. $\Delta \alpha^m$ is an approximation of the vector-jacobian product, which we are inspired by the work of pFedHN[5] and LOOKAHEAD[14]. Given the difference $\Delta \alpha^m$, the update rules for $\psi$ and $e^m$ are:

For $\psi$:

$\Delta \psi = (\nabla_\psi \alpha^m)^{T} \Delta \alpha^m$

$\psi \leftarrow \psi + \eta \frac{1}{M} \sum_m g_m (\nabla_\psi \alpha^m)^{T} \Delta \alpha^m$, using the weights $g_m$ (defined in L56)

For $e^m$:

$\Delta e^m = (\nabla_{e^m} \psi)^{T} (\nabla_\psi \alpha^m)^{T} \Delta \alpha^m$, using the chain rule given $\alpha^m = h_\psi(e^m)$

$e^m \leftarrow e^m + \eta (\nabla_{e^m} \psi)^{T} (\nabla_\psi \alpha^m)^{T} \Delta \alpha^m$

## Reference
[1] pFL-bench: A comprehensive benchmark for personalized federated learning. NeurIPS, 2022.

[2] Federated meta-learning with fast convergence and efficient communication. arXiv preprint, 2018.

[3] Improving federated learning personalization via model agnostic meta learning. arXiv preprint, 2019.

[4] Personalized federated learning with theoretical guarantees: A model-agnostic meta-learning approach. NeurIPS, 2020.

[5] Personalized federated learning using hypernetworks. ICML, 2021.

[6] Asymptotically exact, embarrassingly parallel MCMC. arXiv preprint, 2013.

[7] A Bayesian Federated Learning Framework with Online Laplace Approximation, arXiv, 2021.

[8] Overcoming catastrophic forgetting by incremental moment matching. NeurIPS, 2017.

[9] Snip: Single-shot network pruning based on connection sensitivity. ICLR, 2018.

[10] Variational dropout and the local reparameterization trick. NeurIPS, 2015.

[11] Federated learning via posterior averaging: A new perspective and practical algorithms. ICLR, 2021.

[12] Variational bayesian dropout with a hierarchical prior. CVPR, 2019.

[13] Learning both weights and connections for efficient neural network. NeurIPS, 2015.

[14] Lookahead Optimizer: k steps forward, 1 step back. 2019.

[15] Personalized federated learning via variational bayesian inference. ICML, 2022.

[16] Fedpop: A bayesian approach for personalised federated learning. NeurIPS, 2022.

[17] Personalized federated learning with gaussian processes. NeurIPS, 2021.

[18] Federated Bayesian Neural Regression: A Scalable Global Federated Gaussian Process. arXiv preprint, 2022.

[19] Personalized Federated Learning via Amortized Bayesian Meta-Learning. arXiv preprint, 2023.

[20] What do we mean by generalization in federated learning?. ICLR, 2022.

[21] Variational Bayesian dropout: pitfalls and fixes. ICML, 2018.

[22] Dropout as a bayesian approximation: Representing model uncertainty in deep learning. ICML, 2016.

[23] Products and convolutions of Gaussian probability density functions. Tina-Vision Memo, 2003.

---

### Decision · Program_Chairs · 2023-09-21

**Decision:**

Accept (poster)

**Comment:**

The submission was generally well-received by the reviewers. The combination of hypernetworks and variational dropout to target Bayesian federated learning is interesting and non-trivial. The results are also promising.

Key requested changes are: (i) discussion of and comparison to existing Bayesian federated learning techniques, (ii) more careful ablation studies + error bars. The authors have promised to have some of these changes in the next iteration. The AC thus recommended acceptance.